# High-dimensional non-Abelian holonomy in integrated photonics

Youlve Chen[1,2], Yunru Fan[3,4], Gulliver Larsonneur[1], Jinlong Xiang [1], An He[1], Guohuai Wang[2], Xu-Lin Zhang [2] ✉, Guancong Ma [5], Qiang Zhou [3,4], Guangcan Guo[3,4], Yikai Su [1] ✉ & Xuhan Guo [1] ✉

Non-Abelian holonomy is known for the robust holonomic unitary behavior exhibited. The associated non-Abelian geometric phase is a promising approach for implementing topologically protected computation. But its realization in application-abundant platforms has been largely elusive. In particular, the observation of universal high-order matrices is difficult due to challenges from increasing the dimensions of degenerate subspace. Here we realize a high-dimensional non-Abelian holonomic device on an integrated multilayer silicon nitride platform, which is compatible with the complementary-metal-oxide-semiconductor process. High dimensional (up to 6), broadband (>100 nm operating bandwidth), and ultra-compact volume non-Abelian holonomy unitary matrices of arbitrary special orthogonal group are observed, and $M \times N$ linear holonomic computation architecture is experimentally realized through singular value decomposition. Our work provides a paradigm for versatile applications of non-Abelian geometric phase for both classical and quantum realms.

Holonomy, which refers to the scenario that a vector parallel transports around a closed loop and returns to the start point[1], has attracted dramatic attention since the geometric phase can be acquired in this process. In contrast to the dynamic phase, the geometric phase cannot be removed under gauge transformation since it is solely dependent on the evolution path in Hilbert space. One well-known geometric phase is the Pancharatnam-Berry phase[1,2], which is a one-dimensional geometric phase that belongs to the Abelian (commutative) unitary group $U(1)$. Additionally, F. Wilczek and A. Zee pointed out that high-dimensional geometric phase can be generated by holonomic adiabatic evolution in degenerate subspace[3]. This matrix-valued geometric phase belongs to the unitary group $U(m)$ ($m > 1$), which is a non-Abelian group in nature. This non-Abelian geometric phase that possesses the capability of high-dimensional manipulation is regarded as a promising route for topological-protected computation[4]

and logic manipulation[5]. So far, non-Abelian holonomy has been realized in various systems, including cold atom systems[6,7], superconducting circuits[8], acoustic systems[5,9], and photonic systems[10–17]. The non-Abelian holonomic process can generate arbitrary-dimensional, topologically protected, and wavelength-stable matrices, showing strong potential for manipulating photons working in broadband. Yet, despite various proof-of-concept experiments[11–15,17] exhibited, general non-Abelian holonomy is limited to three-dimensional[11] special orthogonal group, while arbitrary high-dimensional holonomy is still out of reach. Challenges are from the increase of degenerate subspace[18] and the complex hopping required for $U(m)$, which hinders the realization of more versatile non-Abelian operations. Therefore, architecture in application-abundant and general fabrication technology-compatible photonic platforms is the next step.

[1]State Key Laboratory of Photonics and Communications, School of Information and Electronic Engineering, Shanghai Jiao Tong University, Shanghai 200240, China. [2]State Key Laboratory of Integrated Optoelectronics, College of Electronic Science and Engineering, Jilin University, Changchun, China. [3]Institute of Fundamental and Frontier Sciences, University of Electronic Science and Technology of China, Chengdu 611731, China. [4]Center for Quantum Internet, Tianfu Jiangxi Laboratory, Chengdu 641419, China. [5]Department of Physics, Hong Kong Baptist University, Kowloon Tong, Hong Kong, China.
✉e-mail: xulin_zhang@jlu.edu.cn; yikaisu@sjtu.edu.cn; guoxuhan@sjtu.edu.cn

Here, we experimentally realize SO($m$) non-Abelian holonomy up to $m = 6$ on an integrated silicon nitride photonics platform. Both classical light and single-photon experiments are performed. We also demonstrate the $M \times N$ matrix architecture for light-based linear holonomic computation. In our design, $m$-fold degenerate space is protected by a waveguide array with chiral symmetry, and the complete SO($m$) group is realized through a series of $m$ non-coaxial rotations by manipulating the holonomy to traverse through the parameter space of different dimensions. The scalability makes the parameter space be extended to infinite dimensions, which contributes to the observation of high dimensional holonomy. Finally, the advantages of high scalability (SO(6) and $M \times N$), large operation bandwidth (>100 nm), and ultra-compact volume (~5 × 2 × 600 μm³ for SO(2), 6 orders of magnitude smaller than the state-of-the-art scheme[11]) of the integrated platform are also demonstrated. The holonomy can be measured through both classical and non-classical light. These results provide a promising solution for optical manipulation in both classical and quantum applications, such as optical computing, optical switch, Thouless pump, and reconfigurable geometric phase.

## Results

### High-dimensional non-Abelian holonomy

We implement non-Abelian holonomy on the integrated photonic platform. Based on coupled-mode theory[19], the dynamics of light propagation in waveguides follow a Schrödinger-like equation: $H(z)|\psi(z)\rangle = i\partial_z|\psi(z)\rangle$, where $|\psi(z)\rangle$ represents the optical state in waveguides. We start with considering a Hamiltonian with chiral symmetry[20]:

$$H(z) = \begin{bmatrix} 0_{M \times M} & \boldsymbol{\kappa}(z) \\ \boldsymbol{\kappa}^{\mathrm{T}}(z) & 0_{(M+m) \times (M+m)} \end{bmatrix}, \tag{1}$$

wherein, all the waveguides are divided into two different groups, i.e. one group with $M$ waveguides and the other one with $(M+m)$ waveguides. $\boldsymbol{\kappa}(z)$ represents the $z$-dependent coupling coefficients between waveguides from different groups. The effective indices of waveguides are set to zero after the gauge transformation. The chiral symmetry protects $m$-fold degenerate states[5], which can be used to construct a $U(m)$ holonomy[11].

Through this holonomy, the initial states in the degenerate subspace evolve to final states as $|\psi_{\mathrm{final}}\rangle = U(\gamma)|\psi_{\mathrm{initial}}\rangle$. Here $U(\gamma) = Pe^{i\oint_\gamma A}$, wherein $P$ is the path ordering, $A_{ij} = i\langle D_i|\partial_k|D_j\rangle$ is the Wilczek-Zee connection with $|D_{i(j)}\rangle$ being the $i$th ($j$th) state in the degenerate subspace. This matrix-valued geometric phase belonging to the $U(m)$ group is the well-known Berry-Wilczek-Zee (BWZ) phase[3]. We implement this Hamiltonian on a two-layer silicon-nitride-on-insulator (SNOI) integrated photonic platform. The multilayer silicon nitride device is fabricated by depositing silicon nitride and silica thin film repeatedly. The waveguide patterns are defined by electron beam lithography. They are fabricated on the silicon substrate with a silica buried layer, and more fabrication details can be found in Methods. As shown in Fig. 1a, b, all waveguides have an identical cross-section (800 × 450 nm²), which supports a fundamental transverse electric (TE) mode. The coupling term $\kappa$ is determined by the gap distance and is also wavelength-dependent (Supplementary Figs. 1a, b).

We start from the SO(2) holonomy. We set $M = 1$, $m = 2$ of Eq. (1), where $M$ is the number of central waveguides. As shown in Fig. 1a, b, this system consists of a central waveguide X and surrounding waveguides A, B, C. It supports two degenerate states $|D_1\rangle$ and $|D_2\rangle$ spanning a two-fold degenerate subspace. The holonomic parallel transport of $|D_1\rangle$ and $|D_2\rangle$ is accomplished by an adiabatic cyclic modulation of $\kappa$. As shown in the inset of Fig. 1a, this parameter manifold is isomorphic to a unit 2-sphere defined by $\kappa/|\kappa|$. Following the red path, the parallel transport encloses a solid angle $\theta$ of the 2-sphere. This solid angle leads to a SO(2) transformation of two

degenerate states. The initial and final states in the degenerate subspace evolve as $|\psi_{\mathrm{final}}\rangle = e^{i\theta\sigma_y}|\psi_{\mathrm{initial}}\rangle$, where $\sigma_y$ is the Pauli matrix[21], $|\psi_{\mathrm{initial/final}}\rangle = [|w_B\rangle, |w_C\rangle]$ since the start/end points of coupling coefficients are located at the north pole of the 2-sphere (labeled by a star). The sign of $\theta$ is determined by the rotation direction, and each element of SO(2) can be expressed as $e^{i\theta\sigma_y}$. We simulate $\theta \in [0, \pi/2)$ for different elements of SO(2) through both mathematical calculation and device simulation (Supplementary Figs. 31–32). The mathematical method involves the calculation of the BWZ phase. The device simulation uses 3D finite-difference time-domain (3D-FDTD) full-wave simulation for the electromagnetic field through commercial software Lumerical[22] and Max-optics Studio[23]. The results from these two methods fit well with each other. The lower panel of Fig. 1b shows the simulated light magnitude distribution with $\theta = \pi/12$. Nearly 95% of the power remains in the dark state[11] at the output port (waveguides B and C), indicating that the device length is long enough to achieve an adiabatic transfer of light.

The high-dimensional SO($m$) holonomy which requires $m$-fold degenerate states can be generated through a combination of non-coaxial rotations. The number of rotations required is $m$, and the completeness has been demonstrated by Reck-Zeilinger[24] and William[25]. Leveraging the advantage of the compact footprint and high scalability of the integrated platform, SO($m$) holonomy is now available. We refer to the Hamiltonian in Eq. (1), set $M = m/2$ or $m/2 - 1$ when $m$ is even, and $M = (m-1)/2$ when $m$ is odd ($M$ is the number of central waveguides). This system possesses an $m$-fold degenerate subspace. Taking SO(3) as an example ($m$ is odd), it is shown in Fig. 1c that the waveguide system with $M = 1$ and $m = 3$ protects three degenerate states. The initial/final states are input/output from three waveguides labeled with $|1\rangle$, $|2\rangle$, $|3\rangle$. Elements of SO(3) can be generated by three rotations with rotation angles $\theta_1, \theta_2,$ and $\theta_3$. The holonomy travels through four-dimensional parameter space spanned by $\boldsymbol{\kappa} = [\kappa_1, \kappa_2, \kappa_3, \kappa_4]$, as shown in Fig. 1e. For the first rotation, the evolution of $[\kappa_1, \kappa_2, \kappa_3]$ encloses a solid angle $\theta_1$ while $\kappa_4$ keeps zero, which constructs a basic generator of the SO(3) group. After that, $\kappa_2$ keeps zero, while $[\kappa_1, \kappa_3, \kappa_4]$ enclose the second solid angle $\theta_2$, contributing to another non-coaxial rotation. These two rotations have orthogonal rotation axes. Finally, $\kappa_4$ is re-zeroed, and $[\kappa_1, \kappa_2, \kappa_3]$ form the last degree of freedom $\theta_3$. Although the waveguides are discontinuous, the states and $\boldsymbol{\kappa}$ space are continuous.

Figure 1d demonstrates the SO(4) holonomy ($m$ is even), which is the alternate cascade of system with $M = 2$ and system with $M = 1$. Four rotations provide the full six degrees of freedom for SO(4). For the first rotation with $M = 2$ system, as Fig. 1f shows, the holonomy travels through the six-dimensional $\boldsymbol{\kappa}$ space along the orange path, encircling two solid angles $\theta_1$ and $\theta_2$. These two rotations are in two orthogonal planes. Afterward, the outside sphere is projected to a plane and $\boldsymbol{\kappa}$ space is reduced to five dimensions ($M = 1$ system). The four degenerate states are continuous during this projection. In this scheme with $M = 1$, $\kappa_2$ and $\kappa_5$ keep zero while $[\kappa_1, \kappa_3, \kappa_4]$ evolve along the blue path to enclose the solid angles $\theta_3$. The following two rotations experience similar dimensionality increase or reduction to offer the last three degrees of freedom ($\theta_4, \theta_5,$ and $\theta_6$).

Any higher-order SO($m$) holonomy can be realized through this scheme, as demonstrated in Fig. 1g, h. The $m$-fold degenerate subspace is supported by the waveguide array with chiral symmetry. If $m$ is odd, the holonomy traverses through the $\boldsymbol{\kappa}$ space of $(3m-1)/2$ dimensions. If $m$ is even, the holonomy travels in the $\boldsymbol{\kappa}$ space between $3m/2$ dimensions and $(3m/2-1)$ dimensions through dimensionality increase or reduction. The path of holonomy encircling a series of solid angles in different spaces generates the full $m(m-1)/2$ degrees of rotation freedom of SO($m$). Excellent scalability allows the $\boldsymbol{\kappa}$ space to be extended to infinite dimensions. This matrix-valued geometric phase has a unique broadband advantage compared with the dynamic phase. Although $\kappa$ is sensitive to wavelength[26], the geometric phase $\theta$

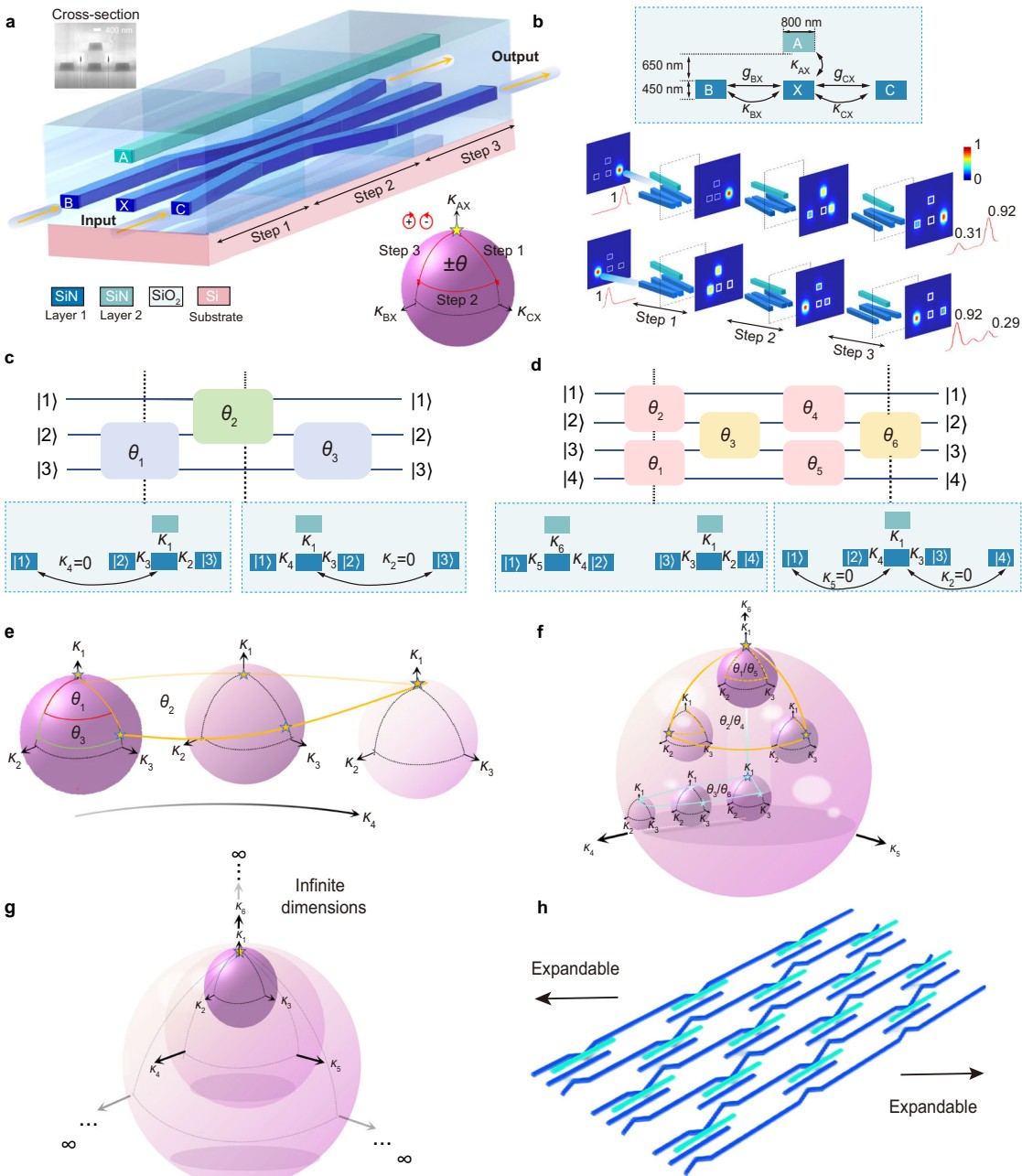

**Fig. 1 | Schematic of high-dimensional non-Abelian holonomy. a** Schematic of four waveguides structure in a two-layer SNOI platform for realizing SO(2) holonomy. The upper inset shows the focused ion beam (FIB) image in a cross-section view, and the lower inset demonstrates the holonomy evolving in the **κ** sphere. For better observation, the coupling coefficient space is normalized as **κ**/|**κ**| to a 2-sphere. The direction of rotation determines the sign of $\theta$. **b** The upper panel shows the details of waveguide parameters in a cross-section view. The lower panel shows an example of FDTD simulation of light magnitude distribution with different inputs when $\theta = \pi/12$. **c** Schematic of SO(3). The system with $M = 1$ and $m = 3$ protects three degenerate eigenstates. The initial/final states are input/output from three waveguides labeled with numbers. Three optional $\theta$ of two types of non-

coaxial rotations generate a complete set of SO(3) group. The panels below illustrate the cross-sections of two non-coaxial rotations, as well as **κ** = [$\kappa_1, \kappa_2, \kappa_3, \kappa_4$] for Fig. 1e. **d** Schematic of SO(4). The complete SO(4) group can be generated through the cascade of $M = 2$ system (pink blocks) and $M = 1$ system (orange blocks), providing full six degrees of freedom for SO(4). **e** SO(3) holonomy travels through four-dimensional **κ** space. **κ** and **θ** correspond to Fig. 1c. **f** SO(4) holonomy travels through six-dimensional **κ** space. **κ** and **θ** correspond to Fig. 1d. The orange dashed line represents the projection of the orange line (six-dimensional curve) onto the inner sphere (three-dimensional), encircling the solid angle $\theta_1$ or $\theta_5$. **g** Infinite dimensions of **κ** space for SO(m) scheme in Fig. 1h. **h** Expandable schematic for SO(m) holonomy.

depends on the quotient of different coupling coefficients, which can largely eliminate the sensitivity to wavelength (see Supplementary Note 2).

## Experiment result of SO(m) holonomy

We perform both classical-light experiments and quantum experiments. The classical experiments illustrate the broadband

characteristic, while the single-photon experiment demonstrates the realization of quantum holonomy.

In classical measurement, a wideband laser is utilized as a light source, and an optical spectrum analyzer (OSA) is used to obtain the transmission spectra as well as the square of elements in the unitary matrix. In quantum measurement, entangled photon pairs are generated through a piece of fiber pigtailed periodically poled LiNbO₃

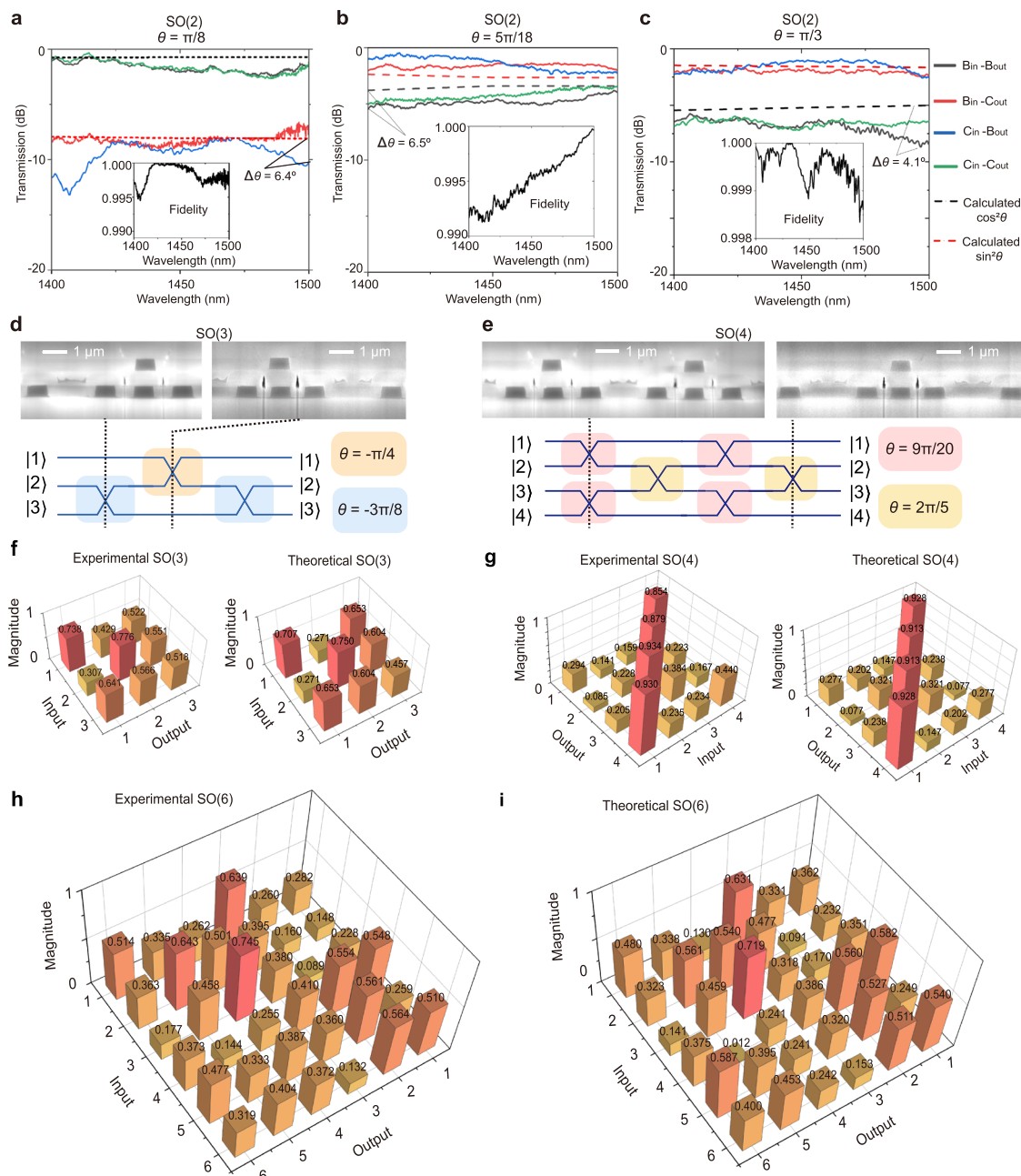

**Fig. 2 | Classical experimental results of high-dimensional SO(m) holonomy.**
**a**–**c** Measured transmission spectra (the solid lines) and theoretical calculation (the dashed lines) of three typical elements of SO(2) with $e^{i\theta\sigma_y}$. The insets show the fidelity-wavelength relation. **a** $\theta=\pi/8$. **b** $\theta=5\pi/18$. **c** $\theta=\pi/3$. **d**, **e** Rotation information of SO(3) and SO(4) and relevant FIB images at specific positions (upper diagram). **f** The measured elements of SO(3) at 1450 nm (left panel) and theoretical prediction (right panel). The parameters are given in Fig. 2d. **g** The measured elements of SO(4) at 1450 nm (left panel) and theoretical prediction (right panel). The parameters are given in Fig. 2e. **h**, **i** The measured elements of SO(6) at 1450 nm and theoretically predicted elements of SO(6).

waveguide[27]. One single photon (at wavelength 1531.72 nm) is injected into the device. The other single photon (at wavelength 1549.34 nm) is injected into a single-mode fiber. Then the two photons are detected by two superconducting nanowire single-photon detectors. A time-to-digital converter is used to record the counting rates and the coincidence events between two photons[27]. The measurement error is calculated through propagating error assuming Poissonian statistics[28–31]. More detailed information can be found in Methods and Supplementary Figs. 33–34. Figure 2a–c shows the measured classical transmission spectra (the solid lines) and theoretical calculation (the dashed lines) of three typical elements of SO(2) with $e^{i\theta\sigma_y}$. These $\theta$ approximately equal to $\pi/8$ (< $\pi/4$, diagonally dominate), $5\pi/18$ (near $\pi/4$), and $\pi/3$ (> $\pi/4$, anti-diagonally dominate), respectively. The

specific values of the Wilson loop[32] are approximately 1.82, 1.3, and 1.07, respectively (see Supplementary Fig. 7 for the integral of the Wilczek-Zee connection). The conformity between theoretical prediction and experimental results is evaluated as fidelity, which is defined as $F = (1/N)|\mathrm{Tr}(U_t^\dagger U_{exp})|$[33–36] based on Frobenious inner product, where $\mathrm{Tr}(A)$ represents the trace of $A$, $N$ is the dimension of the unitary matrix, $U_t$, $U_{exp}$ are the theoretical and experimental unitary matrix, respectively. We can observe a wide bandwidth of the holonomy of more than 100 nm, with the fidelity ranging 0.995–0.999, 0.991–0.999, and 0.998–0.999 in this 100 nm bandwidth respectively. These results indicate that the devices have relatively high fidelities (mostly > 0.99), wide bandwidth (> 100 nm) and low insertion loss (<2 dB). Their magnitude at 1450 nm, as well as more elements of SO(2)

with different $\theta$, are also experimentally realized and shown in Supplementary Fig. 8. The phase measurement can be found in Supplementary Note 4. For the quantum experimental results of different elements of SO(2), the detection probability, measurement errors, and coincidences are shown in Supplementary Fig. 24.

We experimentally realize a SO(3) holonomy. The rotation angles $\theta_1$, $\theta_2$, and $\theta_3$ are shown in Fig. 2d. The FIB images above show the cross-section of SO(3) samples at specific positions. The classically measured elements of SO(3) at 1450 nm and theoretically predicted ones are shown in Fig. 2f. The measured classical transmission spectra and fidelity-wavelength relation are shown in Supplementary Figs. 18b–e. The working bandwidth is larger than 50 nm with fidelity ranging 0.982-0.994, while the Wilson loop $W = \mathrm{Tr}(U)$ is −0.5, which differs from their dimensionality, demonstrating the non-Abelian characteristic[32]. For the quantum experiments of this SO(3), the detection probability, measurement errors, and coincidences are shown in Supplementary Fig. 25a. We also experimentally realize a diagonally dominant SO(3), with the corresponding rotation angles, classically measured, theoretically predicted matrix elements at 1450 nm, transmission spectra, and fidelity-wavelength relation shown in Supplementary Fig. 9. The working bandwidth is larger than 50 nm with fidelity ranging 0.983-0.993.

We also experimentally realize a SO(4) holonomy. The rotation angles and FIB images are shown in Fig. 2e. As mentioned above, it is composed of four non-coaxial rotations, which is accomplished through dimensionality increase or reduction between the waveguide system $M = 2$ and $M = 1$. The holonomy traverses through the $\boldsymbol{\kappa}$ space with different dimensions, which always protects a four-fold degenerate subspace. The classically measured elements of SO(4) at 1450 nm and theoretically predicted elements are shown in Fig. 2g, with a fidelity of 0.991. The Wilson loop is −1.197, showing the non-Abelian characteristic[32]. The quantum experimental results of the SO(4) holonomy are shown in Supplementary Fig. 25b.

Higher-dimensional holonomy can be realized through this scheme. To show the scalability, we experimentally realize an example of SO(6) as well. The classically measured elements of SO(6) at 1450 nm and theoretically predicted ones are shown in Fig. 2h, i. The measured classical transmission spectra and fidelity-wavelength relation are shown in Supplementary Figs. 19–20. The working bandwidth is larger than 50 nm, and the fidelity is above 0.976 in this wavelength range with the highest fidelity of 0.986. The quantum experimental results of this SO(6) are shown in Supplementary Fig. 26.

In theory, the detected probability corresponds to the absolute square of the elements of SO($m$). Making a comparison between classical and quantum measurements, they fit relatively well in most cases. Some distinctions come from the difference of wavelength, since the quantum source provides the single photons at wavelength 1531.47 nm, while the classical experiments are performed in the wavelength range 1400–1500 nm.

Although the Hamiltonian with chiral symmetry protects two degenerate modes, some undesirable effects will break the degeneracy in actual devices[37] especially when the gaps ($g_{\mathrm{BX}}$, $g_{\mathrm{CX}}$) are narrow. The voids generated through plasma-enhanced chemical vapor deposition (PECVD) can partly suppress these undesirable factors, and the voids are reproducible and can be controlled through different processes (Supplementary Note 2 and Supplementary Figs. 5–6). Besides, fabrication deviation will affect the cross-section of waveguides. We simulate its influence on effective indices and experimentally investigate the influence of varying width on the fidelity of holonomy, which are shown in Supplementary Figs. 23 and 28.

## High-dimensional non-Abelian braiding

The above discussion involves ordinary SO($m$). Now we discuss the limiting case with the angle $\theta = \pi/2$, which is also known as the two-mode braiding[5,12]. It requires the holonomy to evolve along the border

of the first octant in the 2-sphere of $\boldsymbol{\kappa}$ space (Fig. 1a). However, due to the fixed non-zero $\kappa_{\mathrm{AX}}$ in the whole evolution, the angle $\theta$ can approach but never reach the limiting value of $\pi/2$. In order to tackle this special situation, we resort to a scheme as shown in Fig. 3a. It can be viewed as the parallel transport of an isolated waveguide and a three-waveguide stimulated Raman adiabatic passage (STIRAP)[38–42], as shown in Fig. 3b–d. The adiabatic evolution of this zero mode exhibits a power transfer from one outer waveguide to another, while acquiring a geometric phase $\pi$. The classical experimental results of power transfer, phase information, and broadband characteristics as well as simulation results can be found in Supplementary Fig. 17a, b and Supplementary Figs. 10-12. The quantum experimental results of two-mode braiding are shown in Supplementary Fig. 27a.

The permutation of an $m$-mode braiding ($m > 2$) is not commutative. Supplementary Figs. 17c, d shows the experimental results of three-mode braiding with different permutations to demonstrate the non-Abelian characteristic. For high-dimensional braiding, an expandable scheme is shown in Fig. 3f, and the classical experimental results of six-mode braiding are shown in Fig. 3h. To illustrate the broadband characteristic, the classical transmission and maximum crosstalk spectra of a five-mode braiding are presented in Fig. 3e (see Supplementary Fig. 13 for the complete transmission spectra). We choose the output port with the maximum crosstalk for each input and draw them in Fig. 3e. For instance, when the input is from port 1, it is clear that the output from port 1 exhibits the maximum crosstalk. Hence, we attach this curve in Fig. 3e. The processes with input from $|2\rangle$ to $|5\rangle$ experience only one braiding operation, where the crosstalk is around -20 dB with a 2.5 dB insertion loss within the wavelength range from 1300 nm to 1500 nm. It is worth noting that even when the input from port $|1\rangle$ experiences five braiding operations, a crosstalk around -10 dB can still be maintained. The quantum experimental results of this five-mode braiding are shown in Supplementary Fig. 27b. The six-mode non-Abelian braiding is also experimentally realized in classical, and the measured magnitude and transmission spectra are shown in Fig. 3h and Supplementary Fig. 14, respectively. These results indicate the excellent scalability and broadband characteristics of the integrated platform in generating non-Abelian holonomy-induced unitary matrices.

## Extended functionality: universal linear transformation

Universal linear transformation $M \times N$ matrices play a very important role in classical optical computation and optical neural networks[33,43,44]. However, schemes based on MZI mesh[43] or variable optical attenuator[44] have a narrow bandwidth. Here we realize $M \times N$ matrices based on singular value decomposition (SVD)[45]:

$$W = UDV^{\dagger} \tag{2}$$

In our scheme, all building blocks ($U$, $D$, $V$) of SVD are realized through the holonomy mentioned above. Two rotations $U$ and $V$ are realized by SO($M$), SO($N$) respectively. The rectangular diagonal matrix $D$ can also be realized through a series of SO(2). This scheme can be viewed as the cascading of SO matrices through dimensionality increase or reduction.

The holonomic structure to realize a $2 \times 3$ matrix is shown in Fig. 4a, b. For higher-dimensional $M \times N$ matrices, a $5 \times 2$ matrix is also realized and shown in Fig. 4c, d. Their mathematical models, including rotation information of special orthogonal groups are shown in Fig. 4c. The measured elements at 1450 nm and theoretically predicted ones are shown in Fig. 4d. The fidelity of the $5 \times 2$ matrix is measured to be 0.988. These $M \times N$ matrices are conducive to future applicable scenarios, such as classical optical computation, and broadband optical neural networks[33,43,44].

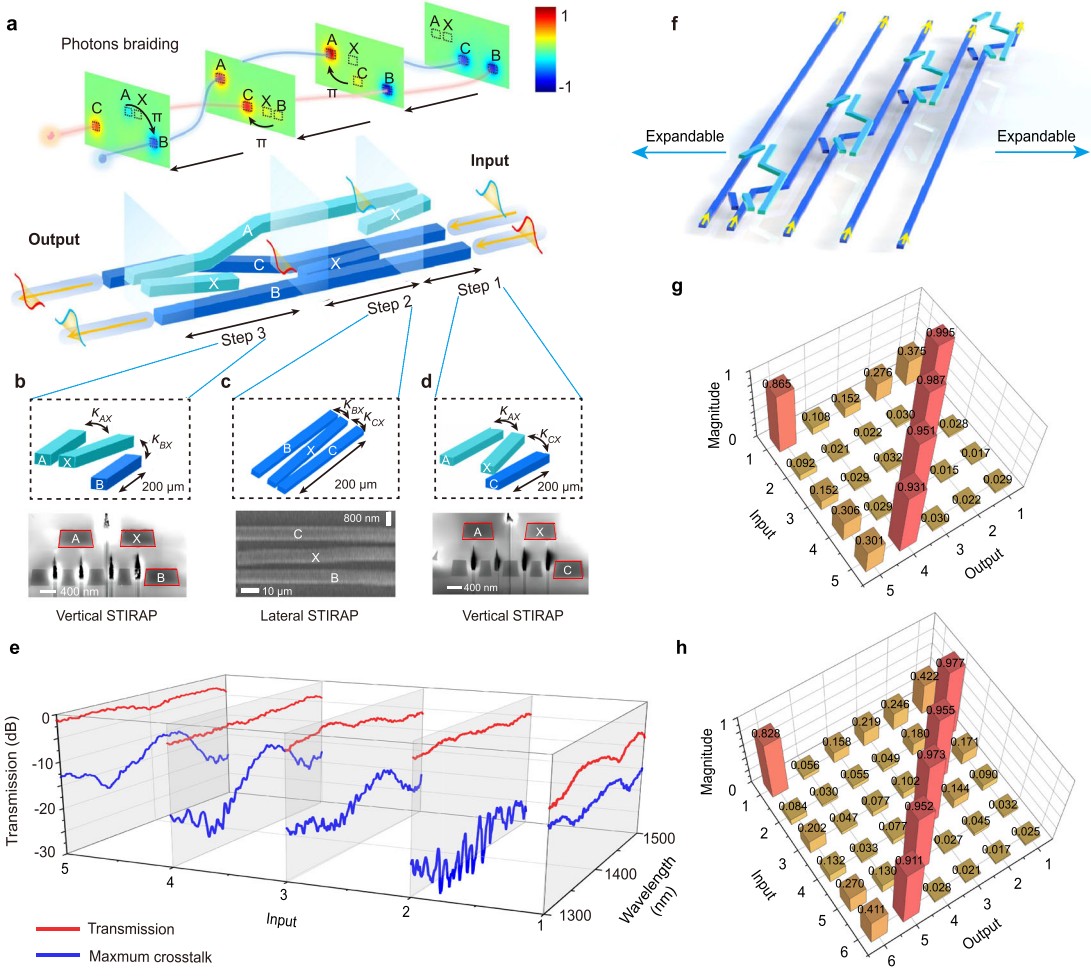

**Fig. 3 | Structure and classical experimental results of high-dimensional non-Abelian braiding. a** Structure diagram of two-mode braiding in a two-layer integrated photonic platform. The inset shows simulated light evolution in three steps (vertical-lateral-vertical STIRAP), where evident power transfers and geometric phase acquired can be observed. **b** Structure diagram of vertical STIRAP and its FIB graph of a cross-section at step 3. **c** Structure diagram of a lateral STIRAP and its SEM images at step 2. **d** Structure diagram of a vertical STIRAP and its FIB graph of a cross-section at step 1. **e** Measured transmission and maximum crosstalk spectra of five-mode non-Abelian braiding. **f** The expandable schematic for high-dimensional braiding. **g** The measured matrix elements of five-mode braiding. **h** The measured matrix elements of six-mode braiding.

## Discussion

We have realized high-dimensional and broadband non-Abelian holonomy with high scalability in an integrated photonic platform. The holonomy is achieved by engineering the coupling coefficients between waveguides placed on two layers. High-order unitary matrices of SO(6) as well as $M \times N$ matrices have been demonstrated. We have also achieved the parallel transport of STIRAP and individual waveguide to realize non-Abelian braiding, up to six-mode braiding with an over 100 nm operation bandwidth. This topologically protected and robust high-dimensional geometric phase with an ultra-compact footprint, large bandwidth, and high scalability will enable us to move towards large-scale integrated functional photonic devices of high-dimensional optical manipulation, such as the spatial optical switch, Thouless pump, large-scale matrices for optical neural networks, and optical computation. Furthermore, how to realize complete unitary group, arbitrary $M \times N$ matrices, and reconfigurable non-Abelian holonomy to enhance the manipulation of photons is a valuable future field to explore. The proposed CMOS-compatible integrated photonic platform may reveal more non-Abelian physics and lead to next-generation on-chip non-Abelian devices for practical applications.

## Methods

### Device fabrication

The fabrication process of a two-layer silicon nitride device is compatible with CMOS technology. The devices were fabricated on a 500-μm-thick silicon substrate wafer with a 3-μm-thick buried oxide layer. For the first layer, a 450-nm-thick silicon nitride film was deposited by PECVD at a deposition rate of around 38 nm/min. The PECVD gas included $SiH_4$, $NH_3$, and $N_2$. Then photoresist (AR-P 6200.13) was spun at a speed of 4000 r/min to have a thickness of 400 nm on the chip, and baked at a temperature of 180 °C for 2 mins. The designed patterns were defined by electronic beam lithography (EBL, Vistec EBPG-5200/JBX-9500FS), and the developing uses methyl isobutyl ketone (MIBK, developing for 75 secs) and isopropanol (IPA, developing for 60 secs). Then it was fully etched through inductively coupled plasma reactive ion etching (ICP etching, NMC) at an etching rate of around 400 nm/min. The etching gas included $CHF_3$, $O_2$, and $N_2$. After that, the chip was coated with silica as inter-layer dielectrics using PECVD. The deposition gas included $SiH_4$ and $N_2O$. If a flat surface is wanted, very thick $SiO_2$ (such as 3 μm) can be deposited first, then etched to the desired thickness. For the second layer, similar processes were repeated as the first layer, such as depositing 450-nm-thick silicon nitride film, defining

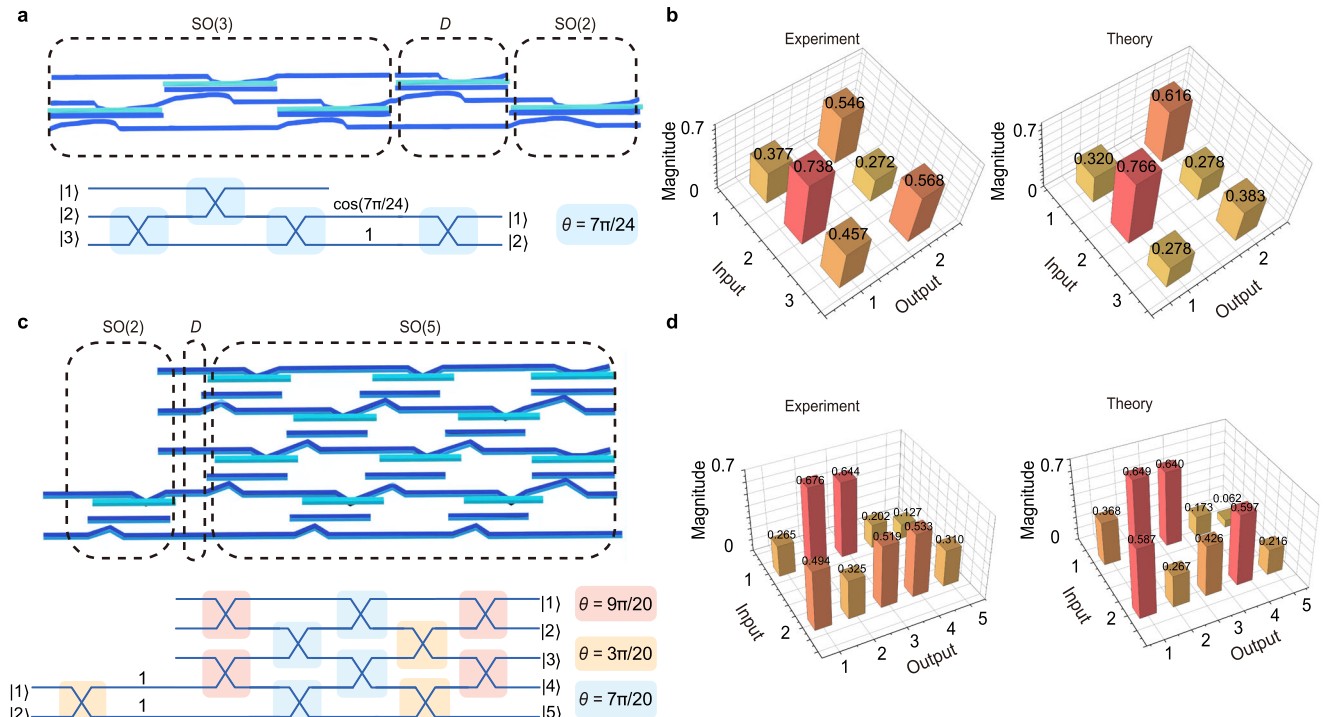

**Fig. 4 | Illustration and experimental results for high-dimensional M×N matrices. a, b** A 2 × 3 matrix. **a** Schematic and mathematical model. **b** Measured and theoretically predicted matrix elements at 1450 nm. **c, d** A 5 × 2 matrix. **c** Schematic and mathematical model. **d** Measured and theoretically predicted matrix elements at 1450 nm.

patterns through EBL, fully ICP etching, etc. Finally, a 3 μm-thick silica cladding was deposited through PECVD. The fabrication process is summarized in Supplementary Fig. 29. After the fabrication, edge couplers need to be exposed for further experiments. Firstly, the double-sided alignment contact UV-lithography (SUSS MA6/MB6) was used to define patterns that protected devices while exposing the region outside the edge couplers. Then, ICP etching was employed to etch dielectrics until the silicon substrate was exposed. After that, ICP deep-silicon etching (NMC) was carried out to etch the silicon substrate about 100 μm. Finally, the die sawing system (SPF-700) was used to slice the chip. This process is summarized in Supplementary Fig. 30.

### Numerical analysis

We used two kinds of simulation methods for mutual confirmation, one is mathematical calculation of the Wilczek-Zee connection integral (Supplementary Equations (21)–(24)) through MATLAB[46], and the other is 3D FDTD[22,23] simulation for the whole device. The coupling coefficients between two identical waveguides are calculated through Mode solutions that $\kappa = (n_s - n_a)/2$[47], where $n_s$, $n_a$ are effective index of symmetry and anti-symmetry supermodes, respectively. The mesh order is set as 3 for a trade-off between accuracy and simulation time. The refractive index of silicon nitride and silica are set as $n_{SiN} = 2.03$ and $n_{SiO2} = 1.47$ for better access to actual fabrication conditions. Some simulation results are shown in Supplementary Figs. 31–32. The two simulations fit well in most cases, the slight deviation comes from some undesirable effects in actual devices such as non-perfect degeneracy.

### Measurement setup

**Classical experiment.** The light is emitted from a wideband source (Wuhan Leicheng Photonics Technology Co., Ltd.), which supports optical wavelength from 1150 nm to 1680 nm. Then it propagates through a polarization controller and is coupled in/out to the integrated photonic chip. Edge couplers (see Supplementary Fig. 16) are utilized to interface the light between single-mode fiber and silicon nitride waveguide, with a coupling loss is approximately 4 dB per facet. The light output is monitored by a power meter (JW3216, Shanghai Joinwit) and the spectra are analyzed through OSA (AQ6370C, YOKOGAWA). These are summarized in Supplementary Fig. 33.

**Quantum experiment.** Photon pairs are generated through second-harmonic generation (SHG) and spontaneous parametric down-conversion (SPDC) in a piece of pigtailed fiber periodically poled LiNbO₃ (PPLN) waveguide. The wavelength of signal and idler photons are 1531.72 and 1549.34 nm, respectively. More details on the generation of entangled photon pairs can refer to ref. 27. One photon is injected into the device. The other single photon is injected into a piece of single-mode fiber. Then the signal and idler photons are detected by two superconducting nanowire single-photon detectors (SNSPDs, P-CS-6, PHOTEC). A time-to-digital converter (TDC, ID900, ID Quantique) is used to record the counting rates and the coincidence events between the signal and idler photons[27]. These are summarized in Supplementary Fig. 34.

### Data availability

The source data generated in this study are available in the Figshare repository at https://doi.org/10.6084/m9.figshare.28578650.

### Code availability

The codes used in this paper are available from the corresponding authors upon request.

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

## Acknowledgements

X.H.G. acknowledges support by National Key R&D Program of China (Grant No. 2023YFB2804700) and Natural Science Foundation of China (Grant No. 62175151), X.L.Z. acknowledges the support by Natural Science Foundation of China (Grant No. 12374350) and The Young Top-Notch Talent for Ten Thousand Talent Program. G.C.M. acknowledges the support by Hong Kong Research Grants Council (Grant No. RFS2223-2S01, 12301822), and the Hong Kong Baptist University (Grant No. RC-RSRG/23-24/SCI/01 and RC-FCRG/23-24/R2/SCI/12). Y.K.S. acknowledges the Natural Science Foundation of China (Grant No. 62341508) and Shanghai Municipal Science and Technology Major Project; Q.Z. acknowledges the support by Sichuan Science and Technology Program (Grant No. 2022YFSY0062). We also thank the Center for Advanced Electronic Materials and Devices (AEMD) of Shanghai Jiao Tong University (SJTU) and Tianjin H-chip Technology for the support in device fabrication, and Beijing MCF Technology LTD for chemical mechanical polishing (CMP) technology support. We would like to thank Mr. Shijun Qiao, Prof. Xincheng Ji and Ms. Linya Zhang for the helpful discussion in the SiN fabrication process.

## Author contributions

X.H.G. initiated the project. Y.L.C. performed the calculation and simulation. X.H.G., Q.Z., Y.R.F. and Y.L.C. designed the experiments. Y.L.C. fabricated samples. Y.L.C. and Y.R.F. carried out the measurements. X.H.G., Y.L.C., G.C.M., X.L.Z., Y.K.S., G.C.G., J.L.X., A.H., G.H.W. and G.L.

analyzed the results and wrote the manuscript. X.H.G., Q.Z., G.C.G., X.L.Z. and Y.K.S. supervised the project.

## Competing interests

The authors declare no competing interests.
