## [Peer Review File · Nature Communications]

High-dimensional non-Abelian holonomy in integrated photonics

Corresponding Author: Professor Xuhan Guo

Version 0:

Reviewer comments:

Reviewer #1

(Remarks to the Author)

The authors report on an implementation of non-Abelian geometric phases associated with group transformations up to $SO(6)$ in an integrated photonics platform. Their implementation clearly shows the potential for realizing holonomic photonic transformations in miniaturized devices. The authors showed that the transformations can be realized with rather large fidelities, and that the device shows broadband capabilities.

However, I have several problems with the manuscript, beyond its poor English:

1. The Hamiltonian (1) suggests that the device can implement holonomies on photonic Fock states which is not true. Indeed, the experiment uses classical light (i.e. coherent states) rather than nonclassical states such as photon pairs. In fact, the proposed use of metasurfaces to induce losses cannot be described by Eq.(1), as it requires an open-systems approach. Hence, the Hamiltonian (1) is inconsistent with the shown results beyond $SO(6)$.
2. Realizing complex couplings has always been an issue in this field, and the authors are very honest about it, which I appreciate. Typically, one would use detuned waveguides for an effective complex coupling. The proposed use of metasurfaces with complex refractive index seems to be incompatible with the way the coupling coefficients between waveguides are derived. The expression given in the Supplementary Material is the standard expression derived for purely real refractive indices for which an expansion into a complete set of orthogonal modes in the waveguides exists. Simply inserting a complex refractive index will have to be justified. Some time ago [M.Golshani et al., Phys. Rev. Lett. 113, 123903 (2014)] this has been tried in an infinite waveguide array, but also there the theory was not complete.
3. It would help if the authors explained why it would be helpful to implement matrix transformations on the basis of rectangular matrices. The authors mention holonomic quantum computation as a possible application, but this does not make sense as an $M \times N$ matrix does not implement any unitary. What exactly do the authors have in mind?

Some minor remarks: For the Wilson loop, Ref.[31] is quoted. However, quantifying non-Abelian holonomies using the Wilson loop goes back several years before Ref.[31].

In the author contributions statement, one of the listed authors (Guancong Ma) does not appear. What was their contribution to the manuscript?

To conclude, I cannot accept the manuscript in its present form.

Reviewer #2

(Remarks to the Author)

The manuscript "High-dimensional non-Abelian holonomy in integrated photonics" by Chen et al describes a scalable waveguide architecture for realizing higher-dimensional non-Abelian holonomies on a silicon nitride platform. As matrix-valued generalization of the Berry phase, non-Abelian geometric phases offer great promise for robust broadband optical computation for both classical and quantum light. While this idea itself is not new, the paper at hand constitutes a significant

technological advance: The presented multilayer technique allows for longer-range interconnects to link otherwise separate domains of the lattice, thereby overcoming the traditional limitations of planar lithographic fabrication without the need for fully three-dimensional arrangements. The manuscript is clearly written and describes the approach in a structured fashion before presenting convincing experimental results that demonstrate the capabilities of the proposed architecture.

A few questions the authors may wish to address to further improve the manuscript:

What are the confidence intervals/measurement errors of the presented data (e.g. fig 2f-h, 3b,d 4g-h)?

what role do the voids that sometimes form in between the waveguides play? By what magnitude do they reduce coupling, and how reproducible is their formation? If stochastic, what impact does this have on the performance of larger systems? If highly reproducible, can they be induced deliberately to locally suppress undesirable interactions? In any case, some discussion of these structures should take place in the main manuscript.

In the SEM images, the structures seem somewhat coarse in places (which is to be expected, given the small dimensions). How accurately can the effective indices (on-site potentials) of the individual waveguides be maintained (or, conversely, tuned) by varying their cross section?

In conclusion, I recommend publication of the manuscript after minor revisions. The topic is current and of significant interest to the integrated optics community, and, while the theoretical background is not new, the technological advancement is definitely noteworthy.

Version 1:

Reviewer comments:

Reviewer #1

(Remarks to the Author)

The authors have substantially modified their manuscript by ensuring that single heralded photons can be used to show braiding. They have also replied to my queries regarding complex waveguide couplings by replacing the metal/metamaterial between waveguide that would induces losses by periodically bent waveguides. Unfortunately, that idea induces new issues, namely bending losses [Opt. Lett. 14, 1231 (1989)] which are particularly pertinent for single-mode waveguides as employed in the present experiment. Indeed, bent waveguides have been used to induce tunable losses, leading to non-unitary state evolution [Nat. Phot. 13, 883 (2019)]. Hence, the authors have not convinced me that periodically bent waveguides indeed lead to unitary evolutions associated with elements of $U(2)$.

The Schrödinger-like equation quoted in line 66 of the main manuscript does not coincide with Eq.(2) in the Supplementary Material. I would have expected both equations to be equivalent when replacing $t \leftrightarrow z$.

In addition, the presentation regarding readability and linguistic accuracy has not been improved substantially. Hence, I cannot recommend the manuscript for publication in Nature Communications.

Reviewer #2

(Remarks to the Author)

Version 2:

Reviewer comments:

Reviewer #1

(Remarks to the Author)

The authors have amended their manuscript according to my remarks and request. In particular, they have answered my question regarding bending losses in waveguides, and cleaned up some of the linguistic issues. I can now recommend the manuscript for publication.

Response letter

Dear reviewers,

We are grateful for the instructive comments as well as the reviewers' time and efforts in accessing our work. In the following response letter, please find a revision checklist and point-by-point response to the referees' comments. We thank the reviewers for their further efforts in accessing our revised manuscript.

Sincerely Yours,

Xuhan Guo on behalf of all co-authors

Revision checklist

Reviewer #1

Comments	Main manuscript revision	Supplementary Information revision
1	(line 66-90, page 2-3): Revise Hamiltonian (1) into the classical form according to classical experiments. (line 318-327, page 10-11, Fig. 4): Delete the use of metasurface. Update Fig. 4a with new experimental results. Move rectangle matrix to the last part as an extended application.	(Supplementary Figs. 24-27): Add new quantum holonomy experiments. (Supplementary Fig. 23): Delete the use of metasurface.
2	(line 158-172, page 4): Add a new method to realize effective complex coupling through periodic bending based on Floquet theory, which replaces the previous inserting metal method.	(Supplementary Note 4): Derive the new method of effective complex coupling through periodic bending waveguides. (Supplementary Fig. 23): Simulate the complex coupling and introduce it to the holonomy.
3	(line 332-334, page 11): Delete holonomic quantum computation as application; Add some citations about the applications of rectangle matrix in classical optical computation.	
4	(Ref.[31]): Revise the citation about the Wilson loop. (line 432-435, page 13): Revise the author contribution list.	

Reviewer #2

Comments	Main manuscript revision	Supplementary Information revision
1	(labeled in red, page 6-7): Add the measurement error of the experiments. (Extended Data Fig. 6): Add quantum experiments setups.	(Supplementary Figs. 24-27): Add measurement error of quantum-mechanical experiments.

2	(line 256-260, page 7): Discuss the role of voids. Experimentally demonstrate the reproducibility and controllability of voids through different fabrication processes.	(Supplementary Fig. 5): Update the influence of voids. (Supplementary Fig. 6.): Different voids generated through different processes.
3	(line 260-263, page 7): Add the influence of deviation in the cross-section of waveguides, including simulations and experiments.	(Supplementary Fig. 28): Effective index variation with the change of waveguide width. (Supplementary Fig. 29): Experimental results of introducing width mismatch for different SO(2).

Reply to Reviewer 1

Comments from Reviewer 1-1:

Reviewer #1 (Remarks to the Author):

The authors report on an implementation of non-Abelian geometric phases associated with group transformations up to $SO(6)$ in an integrated photonics platform. Their implementation clearly shows the potential for realizing holonomic photonic transformations in miniaturized devices.

The authors showed that the transformations can be realized with rather large fidelities, and that the device shows broadband capabilities.

However, I have several problems with the manuscript, beyond its poor English:

Response:

We thank the reviewer's hard work and instructive comments. We will do our best to address these problems, including a series of new quantum experiments to show the capability of quantum holonomy, a new method for complex coupling as well as theoretical derivation and simulations, and new experiments for rectangle matrices, which are shown in the point-to-point responses that follow. We also polish the text and the corrections are marked in red.

Comments from Reviewer 1-2:

1. The Hamiltonian (1) suggests that the device can implement holonomies on photonic Fock states which is not true. Indeed, the experiment uses classical light (i.e. coherent states) rather than nonclassical states such as photon pairs.

Response:

We agree with the reviewer's instructive comments, and we want to address this problem from two aspects:

1. In the revised main manuscript, we use the classical form of the formulas based on coupled-mode theory, rather than quantum Fock states.

2. To show the device can also be implemented in quantum photonics, we cooperate with the UESTC quantum team to address a series of quantum experiments, which are added in the revised Supplementary Information, conforming to the original Hamiltonian of the quantum form.

1. Revised classical formulas:

Based on coupled-mode theory, the dynamic of light in waveguides follows a Schrödinger-like equation: $-i\partial_z|\psi(z)\rangle=H(z)|\psi(z)\rangle$. Consider the Hamiltonian satisfying the chiral symmetry as follows¹:

$$H(z) = \begin{bmatrix} 0_{M \times M} & \boldsymbol{\kappa}(z) \\ \boldsymbol{\kappa}^T(z) & 0_{(M+m) \times (M+m)} \end{bmatrix}$$

where $\boldsymbol{\kappa}(z)$ represents the z -dependent coupling coefficients between two waveguides coming from two different groups, i.e. one group with M waveguides and the other one with $(M+m)$ waveguides. The effective index of all the waveguides is set to zero after gauge transformation. This Hamiltonian satisfies chiral symmetry, which protects m -fold degenerate states¹, and therefore, this $(2M+m)$ -particle system can be used to construct a $U(m)$ holonomy.

2. New quantum experiments:

We cooperate with the UESTC quantum team to conduct new quantum experiments, and the experiment setup is shown in Fig. R1. Photon pairs are generated through second-harmonic generation (SHG) and spontaneous parametric down-conversion (SPDC) in a piece of fiber pigtailed periodically poled LiNbO₃ (PPLN) waveguide. The wavelength of signal and idler photons is 1531.72 nm and 1549.34 nm, respectively. More details on the generation of correlated/entangled photon pairs can refer to [npj Quantum Inf 7, 123 (2021)]. In the experiment, one single photon is injected into the device, while the other single photon is injected into a piece of single-mode fiber. Such configuration is the same as the braiding measurement in [Nat. Photon. 16, 390-395 (2022)] and U(2) holonomy measurement in [Nat. Phys. 19, 30-34 (2023), U(2) part]. Then the signal and idler photons are detected by two superconducting nanowire single-photon detectors (SNSPDs, P-CS-6, PHOTEC). A time-to-digital converter (TDC, ID900, ID Quantique) is used to record the counting rates and the coincidence events between the signal and idler photons [npj Quantum Inf 7, 123 (2021)]. In this quantum configuration, the Hamiltonian of the system can be written as: $\hat{H} = \sum_{j=1}^{M+m} \sum_{i=1}^M [\kappa_{i,j+M} \hat{a}_i \hat{a}_{j+M}^\dagger + \text{h.c.}]$, where \hat{a}_k and \hat{a}_k^\dagger represent the bosonic creation and annihilation operators for spatial mode of the k th waveguide, respectively [Nat. Phys. 19, 30-34 (2023)]. Owing to the limited time, herein we only address single-photon injection to show that our platform can be provided for quantum holonomy. Multi-photon injection scheme as [Nat. Phys. 19, 30-34 (2023), U(3) part] can also be realized after optical packaging, which is our future goal.

Figure R2 shows some quantum experimental results, which are related to the devices used in Fig. 2 and Fig. 3 of the main manuscript, as well as some devices in the Supplementary Information, including SO(2), SO(3), SO(6), and five-mode braiding. The measurement error is calculated through propagating error assuming Poissonian statistics²⁻⁵.

Fig. R1. Setup of quantum experiments. SHG, second-harmonic generation. SPDC, spontaneous parametric down-conversion. PPLN, fiber pigtailed periodically poled LiNbO₃. PC, polarization controller. TDC, time-to-digital converter. SNSPD, superconducting nanowire single-photon detectors. SMP, single-mode fiber. DWDM, dense wavelength division multiplexing. CW, continuous wave. More details of the correlated/entangled photon pairs source can refer to [npj Quantum Inf 7, 123 (2021)].

Detection probability

Output \ Input	1⟩	2⟩
1⟩	$(23.9 \pm 0.3)\%$	$(76.1 \pm 0.6)\%$
2⟩	$(76.7 \pm 0.6)\%$	$(23.3 \pm 0.2)\%$

Coincidences in 10 s

Output \ Input	1⟩	2⟩
1⟩	25071	79803
2⟩	89338	27145

Detection probability

Output \ Input	1⟩	2⟩
1⟩	$(55.4 \pm 0.2)\%$	$(45.6 \pm 0.1)\%$
2⟩	$(47.2 \pm 0.1)\%$	$(52.8 \pm 0.2)\%$

Coincidences in 10 s

Output \ Input	1⟩	2⟩
1⟩	471135	378703
2⟩	388474	433835

Detection probability

Output \ Input	1⟩	2⟩	3⟩
1⟩	$(39.7 \pm 0.3)\%$	$(9.5 \pm 0.6)\%$	$(50.7 \pm 0.1)\%$
2⟩	$(26.6 \pm 0.3)\%$	$(51.4 \pm 0.5)\%$	$(22.0 \pm 0.3)\%$
3⟩	$(53.7 \pm 0.5)\%$	$(18.0 \pm 0.2)\%$	$(28.4 \pm 0.3)\%$

Coincidences in 10 s

Output \ Input	1⟩	2⟩	3⟩
1⟩	66183	15863	84526
2⟩	29930	57778	24761
3⟩	67309	22520	35608

Detection probability

Output \ Input	1⟩	2⟩	3⟩	4⟩	5⟩	6⟩
1⟩	$(1.9 \pm 0.1)\%$	$(1.4 \pm 0.1)\%$	$(8.7 \pm 0.3)\%$	$(2.5 \pm 0.6)\%$	$(3.8 \pm 0.8)\%$	$(2.4 \pm 0.6)\%$
2⟩	$(7.2 \pm 0.3)\%$	$(12.5 \pm 0.4)\%$	$(41.5 \pm 1.0)\%$	$(12.8 \pm 0.4)\%$	$(12.1 \pm 0.4)\%$	$(12.7 \pm 0.4)\%$
3⟩	$(10.9 \pm 0.3)\%$	$(0.32 \pm 0.04)\%$	$(9.6 \pm 0.3)\%$	$(55.0 \pm 1.1)\%$	$(18.8 \pm 0.5)\%$	$(12.7 \pm 0.2)\%$
4⟩	$(24.3 \pm 0.4)\%$	$(16.3 \pm 0.3)\%$	$(45.3 \pm 0.7)\%$	$(6.6 \pm 0.2)\%$	$(1.1 \pm 0.1)\%$	$(6.2 \pm 0.2)\%$
5⟩	$(11.1 \pm 0.3)\%$	$(8.4 \pm 0.3)\%$	$(4.0 \pm 0.2)\%$	$(14.7 \pm 0.4)\%$	$(8.7 \pm 0.3)\%$	$(53.0 \pm 1.1)\%$
6⟩	$(23.6 \pm 0.7)\%$	$(39.5 \pm 1)\%$	$(1.8 \pm 0.1)\%$	$(24.4 \pm 0.7)\%$	$(0.8 \pm 0.1)\%$	$(10.0 \pm 0.4)\%$

Detection probability

Output \ Input	1⟩	2⟩	3⟩	4⟩	5⟩
1⟩	$(8.3 \pm 0.3)\%$	$(3.0 \pm 0.2)\%$	$(2.0 \pm 0.1)\%$	$(1.11 \pm 0.01)\%$	$(86.5 \pm 1.9)\%$
2⟩	$(99.8 \pm 1.5)\%$	$(0.06 \pm 0.02)\%$	$(0.04 \pm 0.01)\%$	$(0.350 \pm 0.003)\%$	$(0.05 \pm 0.02)\%$
3⟩	$(0.02 \pm 0.01)\%$	$(97.8 \pm 1.1)\%$	$(0.2 \pm 0.1)\%$	$(2.41 \pm 0.01)\%$	$(0.06 \pm 0.01)\%$
4⟩	$(0.04 \pm 0.01)\%$	$(0.003 \pm 0.003)\%$	$(97.8 \pm 1.2)\%$	$(1.27 \pm 0.01)\%$	$(2.0 \pm 0.1)\%$
5⟩	$(0.01 \pm 0.01)\%$	$(0.04 \pm 0.01)\%$	$(0.02 \pm 0.01)\%$	$(96.8 \pm 0.6)\%$	$(3.1 \pm 0.1)\%$

Fig. R2. Quantum experimental results of the devices in Fig. 2 and Fig. 3 in the main manuscript, and Supplementary Information. a, SO(2) with $\theta = \pi/3$. b, SO(2) with $\theta = \pi/4$. c, SO(3). The angles of SO(3) are shown in Fig. 2 of the main text. d, SO(6). e, Five-mode braiding.

Main manuscript revision:

(line 66-90, page 2-3):

Holonomic $U(m)$ requires the construction of a m -dimension degenerate space. Our investigation is on photonic platform. Based on coupled-mode theory¹⁹, the optical dynamics in waveguide system is governed by a Schrödinger-like equation: $\hat{H}(z)|\psi(z)\rangle = -i\partial_z|\psi(z)\rangle$, where $|\psi(z)\rangle$ represents vector states in waveguides. To construct a m -dimension degenerate space, consider a Hamiltonian satisfying the chiral symmetry¹⁸, as follows:

$$H(z) = \begin{bmatrix} 0_{M \times M} & \kappa(z) \\ \kappa^T(z) & 0_{(M+m) \times (M+m)} \end{bmatrix} \quad (1)$$

where $\kappa(z)$ represents the z -dependent coupling coefficients between two waveguides coming from two different groups, i.e. one group with M waveguides and the other one with $(M+m)$ waveguides. The effective index of all the waveguides is set to zero after gauge transformation. This Hamiltonian satisfies chiral symmetry, which protects m -fold degenerate states⁵ and therefore, this $(2M+m)$ -particle system can be used to construct a $U(m)$ holonomy.

Through this holonomy, the initial states lying in degenerate subspace evolve to final states as $|\psi_{final}\rangle = U(\gamma)|\psi_{initial}\rangle$. Here $U(\gamma) = \mathbf{P}e^{i\oint \gamma^A}$, wherein $A_{ij} = i\langle D_i | \partial_\kappa | D_j \rangle$ with $|D_{i,j}\rangle$ being the i th, j th eigenstates in the degenerate subspace, \mathbf{P} is the path ordering. This matrix-valued geometric phase belonging to $U(m)$ group is the well-known Berry-Wilczek-Zee (BWZ) phase³. We implement this Hamiltonian on a two-layer silicon-nitride-on-insulator (SNOI) integrated photonic platform. The multilayer silicon nitride waveguides are fabricated through depositing silicon nitride thin film, defining pattern and depositing silica inter-layer thin film in sequence repeatedly on silicon substrate with silica BOX, the details can be found in Method. As shown in Fig. 1a,b, the waveguides are made of silicon nitride and the cladding material is silica. The waveguides have identical cross-sections ($800 \times 450 \text{ nm}^2$) with the same propagation constant, which supports the fundamental transverse electric (TE) mode. κ is determined by the gap distance and is also wavelength-dependent (Supplementary Fig. 1a,b). The initial/final degenerate eigenstates should be designed to lie in input/output waveguides so that $U(\gamma)$ can be measured.

Revision about quantum experiments are summarized as follows:

(line 201-207, page 6): Add new quantum experimental setups.

(line 250-255, page 7): Analyze new quantum experimental results.

(page 19): Figure R1 is added to Extended Data Fig. 6.

Supplementary Information revision:

(Supplementary Figs. 24-27):

The quantum experimental results are added in Supplementary Figs. 24-27.

Comments from Reviewer 1-3:

In fact, the proposed use of metasurfaces to induce losses cannot be described by Eq. (1), as it requires an open-systems approach. Hence, the Hamiltonian (1) is inconsistent with the shown results beyond SO(6).

Response:

We agree with the reviewer's comments. The loss-inducing metasurfaces are not employed to realize SO group of Hamiltonian (1). Instead, here we want to realize an extension of functionality – arbitrary matrix generation using singular value decomposition (SVD). SVD decomposes an arbitrary matrix into two rotational matrices sandwiching a real-diagonal matrix which contains the singular values. The SO holonomies of Hamiltonian (1) are responsible only for the two rotations, whereas the role of loss is to realize the singular values. To enhance the relation between results in Hamiltonian (1) and SVD, in the revised manuscript we will replace the metasurface with a series of SO(2) to realize the diagonal singular matrix of SVD. The following Figs. R3 and R4 show that the matrix D is realized through open systems of a series of SO(2) matrices. Hence, in this new scheme, each building blocks (U, D, V) of SVD are realized through Hamiltonian (1). It can also be viewed as the cascading of SO(m) through ascending or reducing dimensions.

Figure R4 shows the new experiment results of 2×3 matrices. SO(3), SO(2) are responsible for the rotations U and V, and the diagonal matrix D is replaced by a series of SO(2) rather than that based on the metasurface.

Besides, we reorder the main manuscript and put it to the last part, as the extended applications based on previous three parts of SO holonomy.

This universal linear transformation (rectangle matrices $M \times N$) is very useful in optical neural networks⁶⁻⁸ (e.g. fully connected layer, convolution kernel), where most matrices are non-unitary and even non-square. Figure R3 shows two fundamental related works. Figure R8 shows more large-scale versatile applications, which we will discuss in detail in the response of comments 1-5.

Fig. R3| Some related fundamental works of $M \times N$ matrices through singular value decomposition scheme (SVD). a, Adapt from [Optica 5, 1623-1631, (2018)]. b, Adapt from [Photon. Res. 1, 1–15 (2013)].

Fig. R4| New experiment results of $M \times N$ matrices through singular value decomposition scheme

(SVD). **a,c**, Schematic and mathematical model. **b,d**, Measured and theoretical matrix elements at 1450 nm.

Main manuscript revision:

(line 318-327, page 10-11):

Extended functionality: universal linear transformation

Universal linear transformation $M \times N$ matrices play a very important role in classical optical computation and optical neural networks⁵⁸⁻⁶⁰. However, schemes based on MZI mesh⁵⁸ or variable optical attenuator⁵⁹ have a narrow bandwidth. Here we realize $M \times N$ matrices based on singular value decomposition SVD²⁰:

$$W = UDV^\dagger \quad (3)$$

all building blocks (U, D, V) of SVD are realized through the holonomy mentioned above. Two rotations U and V are realized by $SO(M)$, $SO(N)$ respectively. The real-diagonal matrix D can also be realized through a series of $SO(2)$. This scheme can be viewed as the cascading of SO matrices through ascending or reducing dimensions.

(Fig. 4, page 11):

We have replaced Fig. 4a with new experimental results and moved it to the last part.

Fig. 4 | Illustration and experimental results for high-dimensional $M \times N$ matrices. **a-b**, A 2×3 matrix. **a**, Schematic and mathematical model. **b**, Measured and theoretically predicted matrix elements at 1450 nm. **c-d**, A 5×2 matrix. **c**, Schematic and mathematical model. **d**, Measured and theoretically predicted matrix elements at 1450 nm.

Supplementary Information revision:

(Supplementary Information Fig. 9)

We have deleted the metasurface part in Supplementary Fig. 9e-f.

Comments from Reviewer 1-4:

2. Realizing complex couplings has always been an issue in this field, and the authors are very honest about it, which I appreciate. Typically, one would use detuned waveguides for an effective complex coupling. The proposed use of metasurfaces with complex refractive index seems to be incompatible with the way the coupling coefficients between waveguides are derived.

The expression given in the Supplementary Information is the standard expression derived for purely real refractive indices for which an expansion into a complete set of orthogonal modes in the waveguides exists. Simply inserting a complex refractive index will have to be justified.

Some time ago [M. Golshani et al., Phys. Rev. Lett. 113, 123903 (2014)] this has been tried in an infinite waveguide array, but also there the theory was not complete.

Response:

We thank the reviewer's instructive comments. We want to politely explain that our previous method for complex coupling is inserting lossy metal between waveguides, rather than metasurface. Several related works also use the third sandwiched lossy waveguides to introduce effective complex coupling, such as anti-PT symmetry⁹⁻¹² shown in Fig. R5b. Besides, the mentioned paper [Phys. Rev. Lett. 113, 123903 (2014)] also gives the valid derivation of coupled-mode-theory in complex refractive indices waveguides. So in the first part, we give a detailed derivation of coupled-mode-theory with complex permittivity perturbation.

However, we must admit these methods relying on the lossy system have some limits: the complex coupling is nonreciprocal ($\kappa_{AB} = \kappa_{BA} \neq \kappa_{BA}^\dagger$) and the system is non-Hermitian, for which we should balance the loss to keep degeneracy. Inspired by the reviewer's mentioned method "detuned waveguide", we find periodic bending waveguides can introduce artificial gauge field to realize effective complex couplings¹³⁻²¹, which may be a better method. It is associated with the well-known Floquet theory and Peierls phase factor^{22,23}. This coupling coefficient is reciprocal ($\kappa_{AB} = \kappa_{BA}^\dagger$) and the system is Hermitian. We will show a series of simulation results in the second part and replace the previous metal method.

1. Explanation of previous inserting metal method

For Fig. R5a, we first consider the Helmholtz equation when single waveguide 1 or 2 exists:

$$[\nabla_{x,y}^2 - \beta^2 + k_0^2 \varepsilon_{1,2}(x)] \varphi_{1,2}(x, y) = 0, \quad (\text{R2})$$

where $\varphi_{1,2}(x, y)$ is the eigenmode of isolated waveguides 1, 2, β is the propagation constant. Now we consider the Helmholtz equation for the whole system with waveguides 1, 2 and the center metal, the relative permittivity distribution $\varepsilon(x) = \varepsilon_{\text{mr}}(x) + i\varepsilon_{\text{mi}}(x) + \varepsilon_1(x) + \varepsilon_2(x)$, as shown in Fig. R5a:

$$[\nabla_{x,y}^2 + \frac{d^2}{dz^2} + k_0^2 \varepsilon(x)] \varphi(x, y, z) = 0, \quad (\text{R3})$$

where, $\varphi(x, y, z) = a_1(z)e^{-i\beta z}\varphi_1(x, y) + a_2(z)e^{-i\beta z}\varphi_2(x, y)$ (consider the large mismatch of metal). $\varepsilon_{\text{mr}}(x)$, $i\varepsilon_{\text{mi}}(x)$ are the complex relative permittivity of the center metal.

Substrate Eq. (R2) into Eq. (R3), and consider slowly varying envelope approximation, we can get:

$$-2i\beta \sum_{i=1}^2 \frac{da_i(z)}{dz} \varphi_i(x, y) = -\sum_{i=1}^2 k_0^2 [\varepsilon(x) - \varepsilon_i(x)] a_i(z) \varphi_i(x, y) \quad (\text{R4})$$

Consider $\varphi_1(x, y)$ and $\varphi_2(x, y)$ are orthogonal, left multiplying Eq. (R4) by $\varphi_{1,2}(x, y)$ and doing an integration in x and y , we have:

$$i \frac{da_{1,2}(z)}{dz} = -i\gamma a_{1,2}(z) + \kappa a_{2,1}(z), \quad (\text{R5})$$

Where γ is the loss introduced by the overlap between evanescent wave and metal, and the complex coupling κ can be written as:

$$\kappa = \frac{\omega}{4} \iint \varphi_1(x, y) [\varepsilon_{mr}(x) + i\varepsilon_{mi}(x) + \varepsilon_1(x)] \varphi_2^*(x, y) dx dy \quad (R6)$$

Introducing complex coupling coefficients through lossy detuned waveguides is common, such as the realization of anti-PT symmetry^{13–21}. The sandwiched lossy waveguide can bring effective complex coupling for the two outer waveguides as: $\kappa_{eff} = \frac{|\kappa|^2}{\gamma - i\Delta k}$, despite the derivation being different. ([Physical Review A, 96(5), 053845 (2017)] APPENDIX A, Eq. A9-A10).

Fig. R5] Related works of inducing complex coupling coefficients through lossy waveguide.

We also study the reviewer’s cited paper [Phys. Rev. Lett. 113, 123903 (2014)] carefully. They also utilized the complex permittivity perturbation to realize the complex coupling coefficient, whose derivation shows the expansion of orthogonal modes is valid for the complex refractive indices case. The difference is that they use on-site lossy waveguides to introduce complex perturbation. As stated by the reviewer, the theory in this paper is not complete. Therefore, we choose another method by using detuned waveguides to realize the complex coupling.

2. New method: Floquet artificial gauge field introduces complex coupling coefficients

Inspired by the reviewer’s instructive comments, we find that periodic bending waveguides can introduce artificial gauge field to realize effective complex couplings^{13–21}, which is a better method. The theory followed is adapted from Ref.^{13,16,17}.

We start with a two-waveguide system, as the upper diagram of Fig. R6b shows, the axis of the waveguide is periodically bent along the propagation direction z . The bending profile is $x_0(z)$ with the period A and the magnitude A . Assume the overall electric field has a slowly varying envelope: $E(x, y, z) = \varphi(x, y, z)e^{-i\beta z}$, wherein $\varphi(x, y, z)$ is the overall slowly varying optical field that can be expanded into standard orthogonal modes of two waveguides, β is the propagation constant. Consider the Helmholtz equation, we can get:

$$-2i\beta \frac{\partial}{\partial z} \varphi(x, y, z) = - [\nabla_{x,y}^2 + k_0^2 \varepsilon(x, y, z) - \beta^2] \varphi(x, y, z), \quad (R7)$$

Where the $\varepsilon(x, y, z)$ is the relative permittivity with the periodic bending profile $x_0(z)$.

$k_0 = \frac{2\pi}{\lambda}$. After introducing new variables to “straighten” the waveguides:

$$x' = x - x_0(z), \quad y' = y, \quad z' = z \quad (\text{R8})$$

The partial differential in the left term of Eq. (R7) will produce a gauge field because:

$$\frac{\partial}{\partial z'} \varphi(x - x_0(z), y, z) = \frac{\partial}{\partial z'} \varphi(x', y', z') - \frac{\partial}{\partial x'} \varphi(x', y', z') \frac{\partial x_0(z')}{\partial z'} \quad (\text{R9})$$

Substitute Eq. (R8-R9) into Eq. (R7), and introduce the gauge transformation: $\phi(x', y', z') = \varphi(x', y', z') e^{i\beta \frac{dx_0(z')}{dz'} x' + i\frac{\beta}{2} \int_0^{z'} [\frac{dx_0(Z)}{dZ}]^2 dZ}$, we can get¹⁶:

$$\begin{aligned} -i \frac{\partial}{\partial z'} \phi(x', y', z') &= -\frac{1}{2\beta} [\nabla_{x', y'}^2 + k_0^2 \varepsilon(x', y', z') - \beta^2] \phi(x', y', z') + \\ &\quad \beta \frac{d^2 x_0(z')}{dz'^2} x' \phi(x', y', z') \end{aligned} \quad (\text{R10})$$

We can find that bending can induce on-site detuned potential (last term of Eq. (R10)) after we perform the transformation to “straighten” the waveguides. Now we expand the overall $\phi(x', y', z')$ into standard orthogonal modes of two waveguides: $\phi(x', y', z') = a_1(z) \phi_1(x', y') + a_2(z) \phi_2(x', y')$. And the two straight waveguide modes satisfy:

$$[\nabla_{x', y'}^2 + k_0^2 \varepsilon_{1,2}(x', y') - \beta^2] \phi_{1,2}(x', y') = 0, \quad (\text{R11})$$

Subtract Eq. (R11) into Eq. (R10), consider $x' = \frac{g}{2}$ for waveguide 1 and $-\frac{g}{2}$ for waveguide 2 (g is the center-to-center separation between two waveguides, as Fig. R6b shows), we can get the following equations based on coupled-mode analysis¹⁶:

$$\begin{aligned} i \frac{d}{dz} a_1(z') &= -\frac{k_0 g n_{eff}}{2} \frac{d^2 x_0(z')}{dz'^2} a_1(z') + \kappa a_2(z') \\ i \frac{d}{dz} a_2(z') &= \frac{k_0 g n_{eff}}{2} \frac{d^2 x_0(z')}{dz'^2} a_2(z') + \kappa a_1(z') \end{aligned} \quad (\text{R12})$$

wherein β is substituted by $k_0 n_{eff}$, κ is the coupling between straight waveguides expressed as: $\kappa = \frac{\omega}{4} \iint \phi_1(x', y') [\varepsilon(x', y') - \varepsilon_2(x', y')] \phi_2^*(x', y') dx' dy'$, which is a purely real number.

In the limit of $\Lambda \kappa \ll 1$, using a multiple scale asymptotic technology¹⁶, we set:

$a_1(z') = b_1(z') e^{i \int_0^{z'} \frac{k_0 g n_{eff}}{2} \frac{d^2 x_0(Z)}{dZ^2} dZ}$, $a_2(z') = b_2(z') e^{-i \int_0^{z'} \frac{k_0 g n_{eff}}{2} \frac{d^2 x_0(Z)}{dZ^2} dZ}$. Substituting it into Eq. (R12), we can get¹⁶:

$$\begin{aligned} i \frac{d}{dz} b_1(z') &= \kappa_{eff} b_2(z') \\ i \frac{d}{dz} b_2(z') &= \kappa_{eff}^\dagger b_1(z'), \end{aligned} \quad (\text{R13})$$

where $\kappa_{eff} = \frac{\kappa}{\Lambda} \int_0^\Lambda e^{-ik_0 g n_{eff} \frac{dx_0(Z)}{dZ}} dZ = \frac{\kappa}{\Lambda} \int_0^\Lambda [\cos(k_0 g n_{eff} \frac{dx_0(Z)}{dZ}) - i \sin(k_0 g n_{eff} \frac{dx_0(Z)}{dZ})] dZ$
 $= |\kappa_{eff}| e^{i\xi}$. (R14)

Through designing $x_0(z)$, κ_{eff} can be a complex number, and both $|\kappa_{eff}|$ and ξ can be controlled. Take sinusoidal waveguides as an example. As shown in Fig. R6b, $x_0(z)$ is formed by alternatively connecting two half-period cosine functions with different periods:

$$x_0(z) = \begin{cases} A \cos(\frac{2\pi}{\Lambda_1}(z - m\Lambda)), & m\Lambda \leq z < m\Lambda + \frac{\Lambda_1}{2}, \\ -A \cos(\frac{2\pi}{\Lambda_2}(z - m\Lambda - \frac{\Lambda_1}{2})), & m\Lambda + \frac{\Lambda_1}{2} \leq z < (m+1)\Lambda, \end{cases} \quad (m \in \mathbb{Z}).$$

Based on Eq. (R14), κ_{eff} can be written as:

$$\text{Re}[\kappa_{eff}] = \kappa \left[\frac{\Lambda_1}{2\Lambda} J_0\left(\frac{4\pi^2}{\lambda\Lambda_1} g n_{eff} A\right) + \frac{\Lambda_2}{2\Lambda} J_0\left(\frac{4\pi^2}{\lambda\Lambda_2} g n_{eff} A\right) \right] \quad (\text{R15})$$

$$\text{Im}[\kappa_{eff}] = \kappa \left[\frac{\Lambda_1}{2\Lambda} H_0\left(\frac{4\pi^2}{\lambda\Lambda_1} g n_{eff} A\right) - \frac{\Lambda_2}{2\Lambda} H_0\left(\frac{4\pi^2}{\lambda\Lambda_2} g n_{eff} A\right) \right],$$

where, J_0 is the zero-order Bessel function, H_0 is the zero-order Struve function²⁷, k_0 is substituted by $\frac{2\pi}{\lambda}$. We simulate this complex coupling coefficient in a silicon two-waveguide system.

Based on Eq. (R13), the transfer matrix of a directional coupler with complex coupling κ_{eff} can be expressed as (after gauge transformation):

$$\begin{bmatrix} \cos(|\kappa_{eff}|z) & -i \sin(|\kappa_{eff}|z) e^{i\xi} \\ -i \sin(|\kappa_{eff}|z) e^{-i\xi} & \cos(|\kappa_{eff}|z) \end{bmatrix} \quad (\text{R16})$$

If coupling coefficients are pure real ($\xi=0$), it is a conventional directional coupler and the phase between two waveguides is $\pi/2$, as shown in Fig. R6a. After introducing the Floquet gauge field, for example, we choose $A = 0.5 \mu\text{m}$, $\Lambda_1 = 7 \mu\text{m}$, $\Lambda_2 = 13 \mu\text{m}$, $\lambda = 1.6 \mu\text{m}$, $g = 0.8 \mu\text{m}$, $n_{eff} = 2.3847$, in which case κ_{eff} is approximately a purely imaginary number (i.e., $\xi = -\pi/2$), the phase between two waveguides is 0 or π , as shown in Fig. R6b. For more general complex numbers, as Fig. R7c shows, both $|\kappa_{eff}|$ and ξ can be controlled by varying A (the magnitude of $x_0(z)$). Other parameters such as $\Lambda_{1,2}, \lambda, g, n_{eff}$ are fixed. The solid line is calculated through Eq. (R15), and the dots are simulated through FDTD for the observation of the bending directional coupler's coupling length and phase. These two methods fit relatively well with each other. The deviation in large A due to the deviation of paraxial approximation¹³.

Fig. R6] Periodic bending waveguide introduces effective complex coupling and simulations. a, Conventional directional coupler (DC) with purely real coupling, the phase difference between two waveguides is $\pi/2$ no matter which waveguide is injected. **b,** Floquet directional coupler. The upper diagram shows the structure. The axis of the waveguide is periodically modulated by $x_0(z)$ whose period is A and the magnitude is A . The lower diagram shows a simulated example of a Floquet directional coupler. We choose $A = 0.5 \mu\text{m}$, $\Lambda_1 = 7 \mu\text{m}$, $\Lambda_2 = 13 \mu\text{m}$, $\lambda = 1.6 \mu\text{m}$, $g = 0.8 \mu\text{m}$,

$n_{eff} = 2.3847$ (width 500 nm, height 220 nm, silicon waveguide), in which case the coupling coefficient is approximately a purely imaginary number. The phase difference between two waveguides is π or 0. **c**, More general complex κ_{eff} . We change A varies from 0 to $0.6 \mu\text{m}$ and other parameters are fixed based on Fig. R6 (b). $|\kappa_{eff}|$ (black) and ξ (red) change as A varies. The solid line is calculated through Eq. (R15), and the dots are simulated through FDTD for the observation of the bending directional coupler's coupling length and phase.

We then investigate complex coupling in the holonomy, starting with STIRAP, which is an important part of braiding. In the conventional STIRAP with pure real coupling coefficients, the phase difference in two outer waveguides is π if the zero mode is excited (Fig. R7a). While in Floquet STIRAP with pure imaginary coupling coefficients, the phase difference in two outer waveguides is 0 (Fig. R8b, $\kappa_{CX} = -i, \kappa_{BX} = i$ at the center position). The phase difference of 0 is also observed compared with the reference individual bending waveguide. We then simulate two-mode braiding and introduce the complex coupling at Step 2 (Fig. R7c). Step 1 and Step 3 are all straight waveguides, they still evolve with real coupling coefficients, while Step 2 uses bending waveguides to introduce complex coupling. The integral of the Wilzeck-Zee connection can be referred to Supplementary Eq. (50) and shown in Supplementary Fig. 23g-i. The whole evolution can be expressed as: $|B\rangle \rightarrow |C\rangle, |C\rangle \rightarrow |B\rangle$, which belongs to the $U(2)$ group and corresponds to the X gate in quantum logic. In comparison, conventional two-mode braiding expressed as $|B\rangle \rightarrow |C\rangle, |C\rangle \rightarrow -|B\rangle$ belongs to the $SO(2)$ group and corresponds to the Y gate in quantum logic. This simulation illustrates that the introduction of complex coupling can express more general unitary matrices beyond $SO(2)$.

Fig. R7| Introduce effective complex coupling into STIRAP and two-mode braiding. **a**, Conventional STIRAP with pure real coupling coefficients, the phase difference in two outer waveguides is π when the zero mode is excited. **b**, Floquet STIRAP with pure imaginary coupling coefficients, the phase difference in two outer waveguides is 0. The phase difference of 0 is also observed compared with the reference individual bending waveguide at the output position. Parameters are set as $A = 0.5 \mu\text{m}$, $A_1 = 5 \mu\text{m}$, $A_2 = 11 \mu\text{m}$, $\lambda = 1.6 \mu\text{m}$, $n_{eff} = 2.3847$. The center-to-center separation of waveguides C and X varies from $0.6 \mu\text{m}$ to $0.9 \mu\text{m}$. The center-to-center separation of waveguides B and X varies from $0.9 \mu\text{m}$ to $0.6 \mu\text{m}$. **c**. Simulated two-mode braiding and introduce the complex coupling at step 2. Step 1 and Step 3 are all straight waveguides and evolve with real coupling coefficients. The whole evolution can be expressed as: $|B\rangle \rightarrow |C\rangle, |C\rangle \rightarrow |B\rangle$, which belongs to the $U(2)$ group. This simulation illustrates that the introduction of complex coupling can express more general unitary matrices beyond $SO(2)$.

Main manuscript revision:

(line 158-172, page 4):

The further generalization to the U(2) group needs additional degrees of freedom, which can be generated through σ_y , σ_z , and identity matrix I . The basic generator σ_y can be realized through real κ , while σ_z and I require κ to be a complex number. Periodic bending waveguides can introduce the artificial gauge field to realize effective complex couplings^{45–53} based on Floquet theory. The effective complex couplings can be expressed as: $\kappa_{eff} = \frac{\kappa}{\Lambda} \int_0^\Lambda e^{-ik_0 g n_{eff} \frac{dx_0(z)}{dz}} dz$ ^{47,50,51}, where, the axis of the waveguide is periodically bent along the propagation direction z . The bending profile is denoted by $x_0(z)$ with a period Λ and amplitude A . The symbol g is the center-to-center separation between two waveguides, κ is the coupling between straight waveguides, n_{eff} is the effective index, and $k_0 = \frac{2\pi}{\lambda}$.

Especially, if $x_0(z)$ is composed of two alternatively connected sinusoidal functions with different periods Λ_1 and Λ_2 (Supplementary Fig. 23b), κ_{eff} can be expressed as:

$$\begin{aligned} \text{Re}[\kappa_{eff}] &= \kappa \left[\frac{\Lambda_1}{2\Lambda} J_0\left(\frac{4\pi^2}{\lambda\Lambda_1} g n_{eff} A\right) + \frac{\Lambda_2}{2\Lambda} J_0\left(\frac{4\pi^2}{\lambda\Lambda_2} g n_{eff} A\right) \right] \\ \text{Im}[\kappa_{eff}] &= \kappa \left[\frac{\Lambda_1}{2\Lambda} H_0\left(\frac{4\pi^2}{\lambda\Lambda_1} g n_{eff} A\right) - \frac{\Lambda_2}{2\Lambda} H_0\left(\frac{4\pi^2}{\lambda\Lambda_2} g n_{eff} A\right) \right], \end{aligned} \quad (2)$$

where, J_0 is the zero-order Bessel function, H_0 is the zero-order Struve function⁶³. The theoretical derivation and simulations can be referred to Supplementary Note 4 and Supplementary Fig. 23. As shown in Supplementary Fig. 23d-i, we further introduce the complex coupling into the holonomy to simulate a unitary matrix corresponding to the X gate in quantum logic, which belongs to the U(2) group. This simulation illustrates that the introduction of complex coupling can express more general unitary matrices beyond SO(2).

Supplementary Information revision:

(Supplementary Note 4, Supplementary Fig. 23, labeled in blue):

We have added this discussion of artificial gauge field induce complex coupling in Supplementary Note 4 and Supplementary Fig. 23, which replace the previous metal method.

Comments from Reviewer 1-5:

3. It would help if the authors explained why it would be helpful to implement matrix transformations on the basis of rectangular matrices. The authors mention holonomic quantum computation as a possible application, but this does not make sense as an M*N matrix does not implement any unitary. What exactly do the authors have in mind?

Response:

We thank the reviewer's instructive comments. As Fig. R8 shows, the rectangle M×N matrices play a very important role in classical optical computation and optical neural networks, considering many of our experiments are classical-based. The mention of application in “holonomic quantum computation” is because [Nature 586, 207–216 (2020)] (BOX 1) pointed out that quantum machine learning parallels classical DNN and needs the matrix W . Considering rigorousness, now we will delete the mention of “holonomic quantum computation” as a possible application. Because after further investigation we agree with the reviewer that quantum computation generally needs unitary operations, and has rarely demand for M×N matrices.

1. Applications of $M \times N$ matrices in classical optical computation

Classical optical neural networks (ONN) are composed of linear parts (i.e. matrix operations) and nonlinear parts (i.e. activation functions). The weight matrices in fully connected layers or convolution kernels are generally non-unitary or rectangular matrices. Here we will cite some typical works for large-scale ONNs or optical computation through $M \times N$ matrices. As shown in Fig. R8a, [Nature Photonics 11, 441–446 (2017)] uses these matrices to realize deep learning for vowel recognition. In Fig. R8b, [Science 384, 202-209 (2024)] uses these matrices to realize general artificial intelligence. In Fig. R8c, [Nat Commun 15, 5468 (2024).] uses Xbar structure to realize $N \times M$ matrix with high fidelity.

Fig. R8] Related works of $M \times N$ matrices. a, Optical neural networks for vowel recognition. Adapt from [Nature Photonics 11, 441–446 (2017)]. **b**, Optical neural networks for general artificial intelligence. Adapt from [Science 384, 202-209 (2024)]. **c**, $N \times M$ matrix through Xbar structure. [Nat Commun 15, 5468 (2024)].

2. Some views about potential applications in quantum computation

Some papers hold the view that quantum machine learning parallels classical DNN and needs the matrix W (shown in the following). Of course, on account of rigorousness, now we will delete the mention of “holonomic quantum computation” as a possible application.

[Nature 586, 207–216 (2020)]:

- One way to **implement quantum machine learning parallels classical photonic deep neural network accelerators (Box 1 Figure)**: stages of linear waveguide meshes are connected by activation layers, but these activation layers must have strong coherent (reversible) nonlinearities.

[Nature 586, 207–216 (2020)]: **Box 1 Figure | Quantum optical neural network based on programmable photonics.** Such a network, **implementing the matrix operations W_1 , W_2 and W_3 , is fed by single photons** and nonlinear activation (for example, nonlinear materials or atomic nonlinearities). The final state may be measured to complete a quantum computation or passed into a quantum network.

Main manuscript revision:

We delete the mention of “holonomic quantum computation” as a possible application, and cite several papers to show possible applications in classical optical computation:

(line 332-334 page 11)

These $M \times N$ matrices are conducive to future applicable scenarios, such as classical optical computation, and broadband optical neural networks⁵⁸⁻⁶⁰.

Comments from Reviewer 1-6:

Some minor remarks:

For the Wilson loop, Ref.[31] is quoted. However, quantifying non-Abelian holonomies using the Wilson loop goes back several years before Ref.[31].

In the author contributions statement, one of the listed authors (Guancong Ma) does not appear. What was their contribution to the manuscript?

To conclude, I cannot accept the manuscript in its present form.

Response:

We thank the reviewer’s instructive comments. Here, we add a new citation of a paper about the Wilson loop in the early years, that is, [K. G. Wilson, Confinement of quarks. Phys. Rev. D Part. Fields 10, 2445–2459 (1974)].

Prof. Guancong Ma helped analyze the data, discussed the physics, and revised the manuscript. It is an oversight that he was not included in the contribution statement, which is now revised. We also have a fruitful discussion with Mr. Guohuai Wang from Jilin University in the Floquet directional coupler for the complex coupling coefficients. Besides, to fully address the reviewers' concerns in the quantum field, the UESTC quantum team helped us address new quantum experiments. PhD. Yunru Fan, Prof. Qiang Zhou, and Prof. Guangcan Guo make enormous contributions.

We have added them to the author contribution lists as well.

Main manuscript revision:

(line 432-435 page 13)

X.H.G initiated the project. Y.L.C performed the calculation and simulation. X.H.G, Q.Z, Y.R.F, Y.L.C, designed the experiments. Y.L.C. fabricated samples. Y.L.C, Y.R.F. carried out the measurements. X.H.G, Y.L.C, G.C.M, X.L.Z, Y.K.S, G.C.G, J.L.X, A.H, G.H.W and L.G analyzed the results and wrote the manuscript. X.H.G, Q.Z, G.C.G, X.L.Z, Y.K.S. supervised the project.

Reply to Reviewer 2

Comments from Reviewer 2-1:

Reviewer #2 (Remarks to the Author):

The manuscript High-dimensional non-Abelian holonomy in integrated photonics; by Chen et al describes a scalable waveguide architecture for realizing higher-dimensional non-Abelian holonomies on a silicon nitride platform. As matrix-valued generalization of the Berry phase, non-Abelian geometric phases offer great promise for robust broadband optical computation for both classical and quantum light. While this idea itself is not new, the paper at hand constitutes a significant technological advance: The presented multilayer technique allows for longer-range interconnects to link otherwise separate domains of the lattice, thereby overcoming the traditional limitations of planar lithographic fabrication without the need for fully three-dimensional arrangements. The manuscript is clearly written and describes the approach in a structured fashion before presenting convincing experimental results that demonstrate the capabilities of the proposed architecture.

Response:

We are very thankful for the reviewer's hard work and positive comments. We try our best to address these problems and revise the paper. The revision includes a series of new quantum experiments for measurement errors, new fabrication experiments and simulation for voids, and new experiments for fabrication deviation of cross-sections. All the results are shown in the point-to-point responses as follows.

Comments from Reviewer 2-2:

A few questions the authors may wish to address to further improve the manuscript:

What are the confidence intervals/measurement errors of the presented data (e.g. fig 2f-h, 3b,d 4g-h)?

Response:

We thank the reviewer's instructive comments. Our previous experiments are classical-setup-based since the measurement of broadband characteristics is suitable for classical experiments. The confidence intervals/measurement errors are in the region of quantum experiments. Thus, we carry out new quantum experiments to address these problems. We cooperate with the UESTC quantum team, and the experiment setup is shown in Fig. R9. Photon pairs are generated through second-harmonic generation (SHG) and spontaneous parametric down-conversion (SPDC) in a piece of fiber pigtailed periodically poled LiNbO₃ (PPLN) waveguide. The wavelength of signal and idler photons of the photon pairs are 1531.72 and 1549.34 nm, respectively. More details on the generation of correlated/entangled photon pairs can refer to [npj Quantum Inf 7, 123 (2021)]. In the experiment, one single photon is injected into the device, while the other single photon is injected into a piece of single-mode fiber. Such configuration is the same as the braiding measurement in [Nat. Photon. 16, 390-395 (2022)] and U(2) holonomy measurement in [Nat. Phys. 19, 30-34 (2023), U(2) part]. Then the signal and idler photons are detected by two superconducting nanowire single-photon detectors (SNSPDs, P-CS-6, PHOTEC). A time-to-digital converter (TDC, ID900, ID Quantique) is used to record the counting rates and the coincidence events between the signal and idler photons [npj

Fig.R9| Quantum experiments setup. SHG, second-harmonic generation. SPDC, spontaneous parametric down-conversion. PPLN, fiber pigtailed periodically poled LiNbO₃. PC, polarization controller. TDC, time to digital converter. SNSPD, superconducting nanowire single-photon detectors. SMP, single-mode fiber. DWDM, dense wavelength division multiplexing. CW, continuous wave.

Figures R10-R13 show some quantum experimental results, which are related to the devices used in Fig. 2 and Fig. 3 of the main manuscript, as well as some devices in the Supplementary Information, including three SO(2), SO(3), SO(4), SO(6), five-mode braiding. The quantum experiment of six-mode braiding (main manuscript Fig. 3h) is not addressed owing to relatively large loss (Supplementary Fig. 14), which is beyond the capability of the entangled photon pairs source (~1.8 MHz). Thus, we replace it with five-mode braiding and two-mode braiding.

The detected probability and measurement error are shown in Figs. R10-R13. In theory, the detected probability corresponds to the square of magnitude in Fig. 2 and Fig. 3 of the main manuscript. After making a comparison between classical experiments shown in the main manuscript and these new quantum experiments, in most cases, they fit relatively well with each other. Some differences are because the wavelength for quantum and classical experiments is different, since the quantum source provides signal single-photon at wavelength 1531 nm, while the classical experiments observed wavelength from 1400 nm - 1500 nm. Thus, the devices with larger bandwidth such as SO(3) and SO(4) have relatively similar results for quantum and classical experiments. While the device with narrower bandwidth such as SO(6) is not similar for classical experiments (at 1450 nm) and quantum experiments (at 1531 nm). The measurement error is calculated through propagating error assuming Poissonian statistics²⁻⁵. More experimental results are added in the revised Supplementary Information.

Detection probability

Output Input	1⟩	2⟩
1⟩	(23.9±0.3)%	(76.1±0.6)%
2⟩	(76.7±0.6)%	(23.3±0.2)%

Coincidences in 10 s

Output Input	1⟩	2⟩
1⟩	25071	79803
2⟩	89338	27145

Detection probability

Output Input	1⟩	2⟩
1⟩	(55.4±0.2)%	(45.6±0.1)%
2⟩	(47.2±0.1)%	(52.8±0.2)%

Coincidences in 10 s

Output Input	1⟩	2⟩
1⟩	471135	378703
2⟩	388474	433835

Detection probability

Output Input	1⟩	2⟩
1⟩	(82.0±0.5)%	(18.0±0.2)%
2⟩	(31.7±0.2)%	(68.3±0.4)%

Coincidences in 10 s

Output Input	1⟩	2⟩
1⟩	145586	31969
2⟩	55616	120019

Fig. R10. Quantum experimental results of SO(2). a, SO(2) with $\theta = \pi/3$. b, SO(2) with $\theta = \pi/4$. c, SO(2) with $\theta = \pi/8$.

Detection probability

Output Input	1⟩	2⟩	3⟩
1⟩	(39.7±0.3)%	(9.5±0.6)%	(50.7±0.1)%
2⟩	(26.6±0.3)%	(51.4±0.5)%	(22.0±0.3)%
3⟩	(53.7±0.5)%	(18.0±0.2)%	(28.4±0.3)%

Coincidences in 10 s

Output Input	1⟩	2⟩	3⟩
1⟩	66183	15863	84526
2⟩	29930	57778	24761
3⟩	67309	22520	35608

Detection probability

Output Input	1⟩	2⟩	3⟩	4⟩
1⟩	(0.91±0.03)%	(1.27±0.04)%	(8.9±0.1)%	(89.0±0.7)%
2⟩	(5.3±0.1)%	(7.1±0.1)%	(71.8±0.6)%	(15.8±0.2)%
3⟩	(4.4±0.2)%	(80.4±1.3)%	(5.0±0.2)%	(10.2±0.3)%
4⟩	(70.6±0.9)%	(18.8±0.4)%	(1.5±0.1)%	(9.1±0.2)%

Fig. R11. Quantum experimental results of SO(3) and SO(4). The multiple angles of SO(3) and SO(4) are shown in Fig. 2 of the main text. **a**, SO(3). **b**, SO(4).

Fig. R12. Quantum experimental results of SO(6).

Fig. R13. Quantum experimental results of braiding. **a**, Two-mode braiding. **b**, Five-mode braiding.

Main manuscript revision:

(line 201-207, page 6):

In quantum-mechanical measurement, entangled photon pairs are generated through pigtailed periodically poled LiNbO₃ waveguide⁴⁴. One single photon (at wavelength 1531.72 nm) is injected into the device. The other single photon (at wavelength 1549.34 nm) is injected into a single-mode fiber. Then the two photons are detected by two superconducting nanowire single-photon detectors. A time-to-digital converter is used to record the counting rates and the coincidence events between

two photons⁴⁴. The measurement error is calculated through propagating error assuming Poissonian statistics⁵⁴⁻⁵⁷.

(line 222-223, page 6):

For the quantum experimental results of different SO(2), the detection probability, measurement errors, and coincidences are shown in Supplementary Fig. 24.

(line 230-231, page 6-7):

For the quantum experiments of this SO(3), the detection probability, measurement errors, and coincidences are shown in Supplementary Fig. 25a.

(line 242-243, page 7):

The quantum experimental results of this SO(4) are shown in Supplementary Fig. 25b.

(line 249-255, page 7):

The quantum experimental results of this SO(6) are shown in Supplementary Fig. 26.

In theory, the detected probability corresponds to the absolute square of the elements of SO(m). Making a comparison between classical and quantum measurements, they fit relatively well in most cases. Some differences come from the difference of wavelength, since the quantum source provides the signal single-photon at wavelength 1531.47 nm, while the classical experiments observed wavelength from 1400-1500 nm. Thus, the devices with larger bandwidth have more similar results for quantum and classical experiments.

(line 286-287, page 9):

The quantum experimental results of two-mode braiding are shown in Supplementary Fig. 27a.

(line 300-301, page 9):

The quantum experimental results of this five-mode braiding are shown in Supplementary Fig. 27b.

(page 19): Fig. R9 is added to Extended Data Fig. 6.

Supplementary Information revision:

Figures R10-R13 (new quantum experimental results) are added to Supplementary Figs. 24-27.

Comments from Reviewer 2-3:

what role do the voids that sometimes form in between the waveguides play? By what magnitude do they reduce coupling, and how reproducible is their formation? If stochastic, what impact does this have on the performance of larger systems? If highly reproducible, can they be induced deliberately to locally suppress undesirable interactions? In any case, some discussion of these structures should take place in the main manuscript.

Response:

We thank the reviewer's instructive comments. The voids can suppress the degeneracy-broken due to some undesirable factors. And the voids are reproducible and can be induced deliberately to locally suppress undesirable interactions. We will have detailed point-by-point explanations for these problems.

1. Suppress undesirable interactions

Although Hamiltonian with chiral symmetry protects two degenerate modes in math, some undesirable effects will break the degeneracy in actual devices²⁴. As Fig. R14a,b shows, when the gaps (g_{BX} , g_{CX}) are narrow, the n_{eff} of two degenerate modes will bifurcate. The significant increased n_{eff} of $|D_1\rangle$ is because of the self-coupling effect²⁵, that is, high-refractive-index SiN waveguide X has

a high overlap region with $|D_1\rangle$. The decreased n_{eff} of $|D_2\rangle$ is because the undesired κ such as κ_{AB} and κ_{AC} which cannot be negligible. However, voids can be generated in narrow gaps with a high depth-to-width ratio through PECVD, which can suppress these undesirable effects at appropriate positions. As Fig. R14c,d show, the bifurcated n_{eff} of $|D_1\rangle$ and $|D_2\rangle$ merge again when voids are added into gaps. Suppressing the self-coupling effect has the most remarkable contributions, owing to the low refractive index of voids. The n_{eff} of $|D_1\rangle$ decreases around 2.8×10^{-3} when $g_{BX} = g_{CX} = 0.3 \mu\text{m}$ at wavelength 1400 nm. Besides, voids can also slightly decrease κ_{AB} and κ_{AC} but are not significant. Taking $g_{BX} = g_{CX} = 0.3 \mu\text{m}$ as an example, κ_{AB} (or κ_{AC}) can have a 20.7% decrease from $1.45 \times 10^{-3} \mu\text{m}^{-1}$ to $1.15 \times 10^{-3} \mu\text{m}^{-1}$ for voids at appropriate position.

Fig. R14| Non-perfect degeneracy in actual devices. **a,c**, Effective indices of four modes calculated through Lumerical Mode Solutions at wavelength 1400 nm as g_{BX} and g_{CX} change. The inset shows the difference δ between the effective indices of two degenerate modes (D_1 , D_2). **a**, Without voids. **c**, With voids. **b,d**, Effective indices of two degenerate modes (D_1 , D_2) as g_{BX} and g_{CX} change (set $g_{BX} = g_{CX}$). The insets show the undesirable effects. **b**, Without voids. **d**, With voids.

2. Reproducibility and controllability

Voids are reproducible as long as the recipe of PECVD is the same. As Fig. R15b shows, we fabricate waveguide arrays, and each image represents the same condition (e.g. gap, process). Voids have similar appearances in different places under the same condition (e.g. gap, process), illustrating the reproducibility. Besides, the voids' position and size are controllable, if we use different processes. Figure R15a shows three different processes to generate voids in different positions (upper or lower), and different sizes (larger and smaller). We will introduce these processes: Process 2 is a normal

operation, after the photoresist is patterned, ICP etching is utilized in Step 1 to etch silicon nitride. Then after removing the photoresist, silica is deposited in Step 2 through PECVD. Different from process 2, process 3 is over-etching around 200 nm in Step 1. Thus, the voids generated during PECVD in Step 2 have a lower position and larger size. Process 1 is based on process 2. After depositing 650 nm silica through PECVD in Step 2, ICP etching is utilized again in Step 3 to etch silica around 400 nm. Thus, the closure of voids reopens and becomes an inverted triangle shape. Finally, PECVD is utilized again to deposit silica in Step 4, and we can obtain smaller voids. It means voids are controllable and can be induced deliberately. Optimizing the parameters of these processes will generate more different shapes and positions of voids for different needs.

Fig. R15| Different voids generated through different processes. **a**, Different processes can generate different voids. Process 2 is a normal operation, after the photoresist is patterned, ICP etching is utilized in Step 1 to etch silicon nitride. Then after removing the photoresist, silica is deposited in Step 2 through PECVD. Different from process 2, process 3 is over-etching around 200 nm in Step 1. Thus, the voids generated during PECVD in Step 2 have a lower position and larger size. Process 1 is based on process 2. After depositing 650 nm silica through PECVD in Step 2, ICP etching is utilized again in Step 3 to etch silica around 400 nm. Thus, the closure of voids reopens and becomes an inverted triangle shape. Finally, PECVD is utilized again to deposit silica in Step 4, and we can obtain smaller voids. **b**, FIB images of voids generated through different processes and different gaps. Each image represents the same process and gap, in which the array of voids have similar appearances, illustrating the reproducibility.

Main manuscript revision:

(line 256-260, page 7):

Although Hamiltonian with chiral symmetry protects two degenerate modes in math, some undesirable effects will break the degeneracy in actual devices³² especially when the gaps (g_{BX} , g_{CX}) are narrow. The voids generated through PECVD can partly suppress these undesirable factors, and the voids are reproducible and can be controlled through different processes. We will discuss these in Supplementary Note 2 and Supplementary Figs. 5-6.

Supplementary Information revision:

We have updated Supplementary Fig. 5 according to Fig. R14 (analysis of undesirable factors).

We have added Fig. R15 (Different voids generated through different processes) into Supplementary Fig. 6.

Comments from Reviewer 2-4:

In the SEM images, the structures seem somewhat coarse in places (which is to be expected, given the small dimensions). How accurately can the effective indices (on-site potentials) of the individual waveguides be maintained (or, conversely, tuned) by varying their cross section?

Response:

We thank the reviewer's instructive comments. Herein we discuss the influence of fabrication error on the waveguide's cross-section. One of the most significant influences is the deviation in width, which comes from electron beam lithography (EBL), inductively coupled plasma (ICP) etching, etc.

Generally, the fabrication error is ± 50 nm for the deviation of waveguide width, which is possibly the largest error in experiments. As Fig. R16a shows, we simulate the individual waveguide's effective indices as the width changes in ± 50 nm at different wavelengths. Δn_{eff} that range from 0.022 to 0.026 are observed at wavelength from 1300 nm to 1600 nm. Besides, the cross-section is often trapezoidal, and different etching recipes and machines would bring different angles of waveguides (the inset of Fig. R16b, for which we use different recipes and machines). Taking both angle and width into consideration, Fig. R16b shows the simulated effective indices of waveguides at wavelength 1450 nm. $\Delta n_{\text{eff}}=0.030$ is observed, which means the variation of angles will enlarge Δn_{eff} . Thus, it is important to use the same process (recipes) for both two layers to make sure that waveguides have the same angle.

Fig. R16| Effective index variation with the change of waveguide width. a, Simulated the waveguide's effective indices as the width variation in ± 50 nm at wavelength from 1300 nm to 1600 nm. Δn_{eff} that range from 0.022 to 0.026 are observed. **b,** The inset shows the FIB images of waveguides' cross-sections with different recipes and machines. The illustration below shows the simulated effective indices as angle and width vary at wavelength 1450nm.

We then experimentally investigate the influence of cross-section deviation (width deviation) on the holonomy. Generally, the width deviation in the same layer is uniform²⁶, that is, all waveguides generally experience uniform wider/narrower width²⁶ in the same layer. Thus, it is valuable to discuss the width mismatch between the two layers since they experience different batches of process. Here, in fabrication, we deliberately induce width mismatch (± 50 nm) in layer 2 (waveguide A) in the whole holonomy and measure the results of different SO(2) with $\theta = \pi/12, \pi/8, \pi/6, \pi/4$ at wavelength 1300 nm and 1350 nm. As Fig. R17 shows, the fidelity decreases as the width mismatch, which is due to the broken degeneracy. Although in some cases width mismatch remains rather high fidelity, the broadband characteristic is affected.

Fig. R17| Experimental results of deliberately introduce width mismatch in layer 2 of different SO(2). **a**, In fabrication we induce width-mismatch (± 50 nm) between two layers in the whole holonomy. **b-i**, Experimentally measured fidelity, elements' magnitude for different SO(2) at wavelength 1300 nm and 1350 nm as the width of waveguide (layer 2) varies. **b**, $\theta=\pi/12$, at wavelength 1300 nm. **c**, $\theta=\pi/12$, at wavelength 1350 nm. **d**, $\theta=\pi/8$, at wavelength 1300 nm. **e**, $\theta=\pi/8$, at wavelength 1350 nm. **f**, $\theta=\pi/6$, at wavelength 1300 nm. **g**, $\pi/6$, at wavelength 1350 nm. **h**, $\theta=\pi/4$, at wavelength 1300 nm. **i**, $\pi/4$, at wavelength 1350 nm.

Main manuscript revision:

(line 260-263, page 7):

Besides, fabrication deviation will affect the cross-section of waveguides, we simulate its influence on effective indices and experimentally investigate the influence of varying width on the fidelity of holonomy, which are shown in Supplementary Figs. 28-29.

Supplementary Information revision:

(Supplementary Figs. 28-29):

We have added Fig. R16 (Effective index variation with the change of waveguide width) and Fig. R17 (Experimental results of introducing width mismatch for different SO(2)) into Supplementary Figs. 28-29.

Comments from Reviewer 2-5:

In conclusion, I recommend publication of the manuscript after minor revisions. The topic is current and of significant interest to the integrated optics community, and, while the theoretical background is not new, the technological advancement is definitely noteworthy.

Response:

We are very grateful for the reviewers' positive comments, which precisely point out the advantages and disadvantages of our work. These valuable comments improve the quality of this paper. We thank the reviewers' hard work again!

Reference

1. Chen, Z. G., Zhang, R. Y., Chan, C. T. & Ma, G. Classical non-Abelian braiding of acoustic modes. *Nat Phys* **18**, 179–184 (2022).
2. Wang, J. *et al.* Multidimensional quantum entanglement with large-scale integrated optics. *Science* **360**, 285-291 (2018)
3. Harris, N. C. *et al.* Quantum transport simulations in a programmable nanophotonic processor. *Nat Photonics* **11**, 447–452 (2017).
4. Qiang, X. *et al.* Large-scale silicon quantum photonics implementing arbitrary two-qubit processing. *Nat Photonics* **12**, 534–539 (2018).
5. Carolan, J. *et al.* Universal linear optics. *Science* **349**, 711–716 (2015).
6. Shen, Y. *et al.* Deep learning with coherent nanophotonic circuits. *Nat Photonics* **11**, 441–446 (2017).

7. Xu, Z. *et al.* Large-scale photonic chiplet Taichi empowers 160-TOPS/W artificial general intelligence. *Science* **384**, 202–209 (2024).
8. Moralis-Pegios, M., Giamougiannis, G., Tsakyridis, A., Lazovsky, D. & Pleros, N. Perfect linear optics using silicon photonics. *Nat Commun* **15**, 5468 (2024).
9. Yang, F., Liu, Y. C. & You, L. Anti-PT symmetry in dissipatively coupled optical systems. *Phys. Rev. A* **96**, 053845 (2017).
10. Zhang, X. L., Jiang, T. & Chan, C. T. Dynamically encircling an exceptional point in anti-parity-time symmetric systems: asymmetric mode switching for symmetry-broken modes. *Light Sci Appl* **8**, 88 (2019).
11. Fan, H., Chen, J., Zhao, Z., Wen, J. & Huang, Y. P. Antiparity-Time Symmetry in Passive Nanophotonics. *ACS Photonics* **7**, 3035–3041 (2020).
12. Wei, Y. *et al.* Anti-parity-time symmetry enabled on-chip chiral polarizer. *Photonics Res* **10**, 76 (2022).
13. Song, W. *et al.* Dispersionless Coupling among Optical Waveguides by Artificial Gauge Field. *Phys. Rev. Lett.* **129**, 053901 (2022).
14. Rechtsman, M. C. *et al.* Photonic Floquet topological insulators. *Nature* **496**, 196–200 (2013).
15. Fang, K., Yu, Z. & Fan, S. Realizing effective magnetic field for photons by controlling the phase of dynamic modulation. *Nat Photonics* **6**, 782–787 (2012).
16. Longhi, S. Coherent destruction of tunneling in waveguide directional couplers. *Phys. Rev. A* **71**, 065801 (2005).
17. Longhi, S. Self-imaging and modulational instability in an array of periodically curved waveguides. *Opt. Lett.* **30**, 2137–2139 (2005).
18. Yi, X., Zeng, H., Gao, S. & Qiu, C. Design of an ultra-compact low-crosstalk sinusoidal silicon waveguide array for optical phased array. *Opt. Express* **28**, 37505 (2020).
19. Garanovich, I. L., Longhi, S., Sukhorukov, A. A. & Kivshar, Y. S. Light propagation and localization in modulated photonic lattices and waveguides. *Physics Reports* **518**, 1–79 (2018).
20. Longhi, S. *et al.* Observation of dynamic localization in periodically curved waveguide arrays. *Phys. Rev. Lett.* **96**, 243901 (2006).
21. Zeuner, J. M. *et al.* Optical analogues for massless Dirac particles and conical diffraction in one dimension. *Phys. Rev. Lett.* **109**, 023602 (2012).
22. R. Peierls. Zur Theorie des Diamagnetismus von Leitungselektronen. *Z. Physik* **80**, 763–791 (1933).
23. Luttinger, J. M. The Effect of a Magnetic Field on Electrons in a Periodic Potential. *Phys. Rev.* **84**, 814 (1951).
24. Yang, Y. *et al.* Non-Abelian physics in light and sound. *Science* **383**, 844–858 (2024).
25. Pinske, J. & Scheel, S. Symmetry-protected non-Abelian geometric phases in optical waveguides with nonorthogonal modes. *Phys. Rev. A* **105**, 013507 (2022).
26. Sun, L. *et al.* Broadband and Fabrication Tolerant Power Coupling and Mode-Order Conversion Using Thouless Pumping Mechanism. *Laser Photon Rev* **16**, 2200354 (2022).
27. Struve, H, Beitrag zur Theorie der Diffraction an Fernröhren. *Annalen der Physik und Chemie.* **253**: 1008–1016 (1882).

Rebuttal letter

Dear reviewers,

We are grateful for the instructive comments as well as the reviewers' time and efforts in accessing our work. We are pleased to read that Reviewer #2 recommended our work for publication by recognizing the thoroughness and convincingness of our response. Despite this recommendation, we are concerned by Reviewer #1 who raised a concern that is not the central claim of our experimental results but only an outlook of this work while recognizing our substantial modification and improved clarity in the revised manuscript. In the following response letter, please find a revision checklist and point-by-point response to the reviewers' comments. We thank the reviewers for their further efforts in accessing our revised manuscript.

Sincerely Yours,

Xuhan Guo on behalf of all co-authors

Revision checklist

Reviewer #1

Comments	Main manuscript revision	Supplementary Information revision
1	(Main manuscript, line 160): Remove the discussion of U(2).	(Supplementary information): Remove the discussion of U(2).
2	(Main manuscript, lines 67): Remove the negative sign of the equation.	
3	(Main manuscript (marked in red)): Further improve the readability and linguistic accuracy (marked in red).	

Reply to Reviewer 1

Comments from Reviewer 1-1:

The authors have substantially modified their manuscript by ensuring that single heralded photons can be used to show braiding. They have also replied to my queries regarding complex waveguide couplings by replacing the metal metamaterial between waveguide that would induce losses by periodically bent waveguides. Unfortunately, that idea induces new issues, namely bending losses [Opt. Lett.14,1231(1989)] which are particularly pertinent for single-mode waveguides as employed in the present experiment. Indeed, bent waveguides have been used to induce tunable losses, leading to non-unitary state evolution [Nat.Phot.13,883(2019)]. Hence, the authors have not convinced me that periodic bent waveguides indeed lead to unitary evolutions associated with elements of $U(2)$.

Reply: We thank the reviewer's instructive comments. Regarding the complex coupling for the $U(2)$ scheme, we state that it only serves as a potential direction for future research outlook in a few sentences, and it is not even the result of this work. All the theoretical and experimental results in our manuscript are in the framework of $SO(m)$. However, in response to the technical concerns raised, we will remove this discussion from the current manuscript as it is not central to our main claims.

However, we are willing to express our view on the issue raised by the reviewer. The paper [Nat. Photonics 13, 883–887 (2019)] cited by the reviewer uses femtosecond laser written silica waveguides with weak light confinement, where the refractive index contrast between the waveguide and background is ~ 0.0008 . As a result, unfortunately, light propagating in the bent waveguides suffers from large radiation losses. This is however completely different from our outlook scheme in the previous version where the bent waveguides are silicon waveguides, which exhibit strong light confinement due to the high refractive index contrast between silicon and background (~ 2.04). The majority of optical power is confined within the silicon waveguide of the guiding mode, only a negligible amount of optical power exists in the evanescent wave (the bending loss comes from the leaky mode in evanescent wave [Opt. Lett.14,1231(1989)]). Thus, our silicon waveguides can bear a very small bending radius (Fig. R2), which has a negligible bending loss for the complex coupling scenario. A recent paper [Opt. Lett. 47, 226-229 (2022)] reported only a total loss of -0.2 dB in periodically bent silicon waveguides for the whole device. Another recent work [Phys. Rev. Lett. 129, 053901 (2022)] also reported only $-0.4\sim -0.6$ dB insertion loss in periodically bent silicon waveguides for complex coupling (similar to the directional coupler, conforming the unitary behavior). We must emphasize that the complex coupling is not from loss but from the artificial gauge field introduced by periodically bent waveguides. A theory regarding this point can be referred to [Phys. Rev. Lett. 129, 053901 (2022), Supplemental Material]. On the other hand, since **all the waveguides have the same bending profile**, all waveguides share the same loss with no relative loss between them even if the negligible loss is considered. Thus, the unitary behavior is still valid under a global gauge transformation, which is distinct from the non-unitary behavior of the non-Hermitian system in [Nat. Photonics 13, 883–887 (2019)] cited by the reviewer.

We exemplify some papers that can confirm our scheme and adapt them in Fig. R1. That is

why we propose the potential scheme of U(2) on the integrated photonic platform. A comparison of the bend-induced loss between the femtosecond laser written waveguides and silicon waveguides is summarized in Fig. R2. You may find that the bending loss of silicon waveguides is lower than that of the femtosecond laser written waveguides on 2~3 orders. In addition, the length of the silicon device is also shorter than the femtosecond laser written device on 2~3 orders. We also experimentally fabricate a 100 μm silicon periodically bent waveguide (sinusoidal bend), and measured the transmission spectrum. As shown in Fig. R3, the insertion loss of is approximate 0 dB, verifying the bending loss is negligible, which is consistent with the results in Fig. R1. As a result, the silicon device with bent waveguides would exhibit a negligible loss, and therefore our proposal for U(m) is valid. We hope the reviewer could reconsider our manuscript.

Phys. Rev. Lett. 129, 053901 (2022)

Opt. Express 28, 37505-37513 (2020)

Opt. Lett. 47, 226-229 (2022)

Fig. R1| Related works of periodic bent waveguides with ultra-low loss to realize abnormal coupling. (a) Experimental results adapted from [Phys. Rev. Lett. 129, 053901 (2022)]. (b) Experimental results adapted from [Opt. Express 28, 37505-37513 (2020)]. (c) Experimental results adapted from [Opt. Lett. 47, 226-229 (2022)].

Fig. R2| Comparison of the reported bending radius and bending loss of silicon waveguide and femtosecond-laser-written silica waveguide. Owing to the high refractive index contrast of silicon waveguides, silicon waveguides can bear a very small bending radius. The bending loss can be negligible when the radius is larger than 5 μm [Opt. Express 12, 1622-1631 (2004)], which is suitable for the periodic bent waveguides. Reference [5] uses sinusoidal waveguides and we choose the minimum bending radius.

- [1] Dajian Liu, et al. High-Order Adiabatic Elliptical-Microring Filter with an Ultra-Large Free-Spectral-Range. *J. Lightwave Technol.* 39, 5910-5916 (2021)
- [2] Y Tan, D Dai. Silicon microring resonators. *J Optics*. 2018; 20(5): 054004.
- [3] W. Zhang, et al. Robust, Compact Microring Resonator Based on Optimized N-Adjustable Curvature, 2024 Conference on Lasers and Electro-Optics Pacific Rim (CLEO-PR), Incheon, Korea, Republic of, 2024, pp. 1-2.
- [4] W. Wang et al., Broadband Mid-Infrared Frequency Comb Generation in a Large-Cross-Section Silicon Microresonator, *IEEE Photonics Journal*, vol. 15, no. 3, pp. 1-6, June 2023, Art no. 6601406,
- [5] T. Eichelkraut, S. Weimann, S. Stützer, S. Nolte, and A. Szameit, "Radiation-loss management in modulated waveguides," *Opt. Lett.* 39, 6831-6834 (2014)
- [6] Klauck, F., Teuber, L., Ornigotti, M. et al. Observation of PT-symmetric quantum interference. *Nat. Photonics* 13, 883–887 (2019).
- [7] Lee, T., Sun, Q., Beresna, M. et al. Low bend loss femtosecond laser written waveguides exploiting integrated microcrack. *Sci Rep* 11, 23770 (2021).
- [8] Welm M. Pätzold, Ayhan Demircan, and Uwe Morgner, "Low-loss curved waveguides in polymers written with a femtosecond laser," *Opt. Express* 25, 263-270 (2017)

Fig. R3| The insertion loss of a 100 μm silicon periodically bent waveguide. The transmission spectrum of 100 μm silicon sinusoidal waveguide is normalized by the transmission spectrum of a 100 μm straight waveguide and grating coupler. We can observe that insertion loss of the bending waveguide is around 0 dB, verifying that the bending loss is negligible. This conclusion is consistent with the papers listed in Fig. R1.

Revision:

(Main manuscript, line 160):

We have removed the discussion of U(2).

(Supplementary):

We have removed the discussion of U(2).

Comments from Reviewer 1-2:

The Schrödinger-like equation quoted in line 66 of the main manuscript does not coincide with Eq.(2) in the Supplementary Material. I would have expected both equations to be equivalent when replacing $t \leftrightarrow z$.

Reply:

We thank the reviewer's very careful check. We have removed the negative sign of the equation in line 66 of the main manuscript.

Revision:

(Main manuscript, line 65-68):

We implement non-Abelian holonomy on the integrated photonic platform. Based on coupled-mode theory, the dynamics of light propagation in waveguides follow a Schrödinger-like equation: $\hat{H}(z)|\psi(z)\rangle = i\partial_z|\psi(z)\rangle$, where $|\psi(z)\rangle$ represents the optical state in waveguides.

Comments from Reviewer 1-3:

In addition, the presentation regarding readability and linguistic accuracy has not been improved substantially. Hence, I cannot recommend the manuscript for publication in Nature Communications.

Reply: We thank the reviewer's instructive comments. We have made further revisions to improve the readability and linguistic accuracy. A part of the revisions is listed in the following, others are marked in red in the main manuscript.

Revision:

(Main manuscript, line 87-94):

We start from the SO(2) holonomy. We set $M = 1$, $m = 2$ of equation (1), where M is the number of central waveguides. This system consists of a central waveguide X and surrounding waveguides A, B, C. It supports two degenerate states $|D_1\rangle$ and $|D_2\rangle$ spanning a two-fold degenerate subspace. The holonomic parallel transport of $|D_1\rangle$ and $|D_2\rangle$ is accomplished by an adiabatic cyclic modulation of κ . As shown in the inset of Fig. 1a, this parameter manifold is isomorphic to a unit 2-sphere defined by $\kappa/|\kappa|$. Following the red path, the parallel transport encloses a solid angle θ of the 2-sphere. This solid angle leads to a SO(2) transformation of two degenerate states.

(Main manuscript, line 95-101):

$|\psi_{\text{initial/final}}\rangle = [|w_B\rangle, |w_C\rangle]$ since the start/end points of coupling coefficients are located at the north pole of the 2-sphere (labeled by a star). The sign of θ is determined by the rotation direction, and each element of SO(2) can be expressed as $e^{i\theta\sigma_y}$. We simulate $\theta \in [0, \pi/2)$ for different elements of SO(2) through both mathematical calculation and device simulation (Extended Data Figs. 3-4). The mathematical method involves the calculation of the BWZ phase. The device simulation uses 3D finite-difference time-domain (3D-FDTD) full-wave simulation for the electromagnetic field through commercial software Lumerical²⁷ and Maxoptics Studio⁴³.

(Main manuscript, line 114):

Elements of SO(3) can be generated by three rotations with rotation angles $\theta_1, \theta_2, \theta_3$. The holonomy travels through four-dimensional parameter space spanned by $\kappa = [\kappa_1, \kappa_2, \kappa_3, \kappa_4], \dots$

(Main manuscript, line 117):

\dots , while $[\kappa_2, \kappa_3, \kappa_4]$ enclose the second solid angle θ_2 , contributing to another non-coaxial rotation. These two rotations have orthogonal rotation axes.

(Main manuscript, line 129):

The following two rotations experience similar dimensionality increase or reduction to offer the last three degrees of freedom (θ_4, θ_5 , and θ_6).

(Main manuscript, line 207):

...it is composed of four non-conaxial rotations, which is accomplished through **dimensionality increase or reduction** between the waveguide system...

Reply to Reviewer 2

Comments from Reviewer 2-1:

The authors have satisfactorily addressed my comments.

Reply:

We are very thankful for the reviewer's very positive comments.